# Harnessing Hard Mixed Samples with Decoupled Regularizer

**Zicheng Liu**[1,2,*]   **Siyuan Li**[1,2,*]   **Ge Wang**[1,2]   **Chen Tan**[1,2]
**Lirong Wu**[1,2]   **Stan Z. Li**[2,†]
AI Lab, Research Center for Industries of the Future, Hangzhou, China;
[1]Zhejiang University;   [2]Westlake University;
{liuzicheng; lisiyuan; wangge; tanchen; lirongwu; stan.zq.li}
@westlake.edu.cn

## Abstract

Mixup is an efficient data augmentation approach that improves the generalization of neural networks by smoothing the decision boundary with mixed data. Recently, *dynamic* mixup methods have improved previous *static* policies effectively (*e.g.*, linear interpolation) by maximizing target-related salient regions in mixed samples, but excessive additional time costs are not acceptable. These additional computational overheads mainly come from optimizing the mixed samples according to the mixed labels. However, we found that the extra optimizing step may be redundant because label-mismatched mixed samples are informative hard mixed samples for deep models to localize discriminative features. In this paper, we thus are not trying to propose a more complicated *dynamic* mixup policy but rather an efficient mixup objective function with a decoupled regularizer named Decoupled Mixup (DM). The primary effect is that DM can adaptively utilize those hard mixed samples to mine discriminative features without losing the original smoothness of mixup. As a result, DM enables *static* mixup methods to achieve comparable or even exceed the performance of *dynamic* methods without any extra computation. This also leads to an interesting objective design problem for mixup training that we need to focus on both smoothing the decision boundaries and identifying discriminative features. Extensive experiments on supervised and semi-supervised learning benchmarks across seven datasets validate the effectiveness of DM as a plug-and-play module. Source code and models are available at https://github.com/Westlake-AI/openmixup.

## 1   Introduction

Deep Learning has become the bedrock of modern AI for many tasks in machine learning [3] such as computer vision [19, 18], natural language processing [12]. Using a large number of learnable parameters, deep neural networks (DNNs) can recognize subtle dependencies in large training datasets to be later leveraged to perform accurate predictions on unseen data. However, models might overfit the training set without constraints or enough data [53]. To this

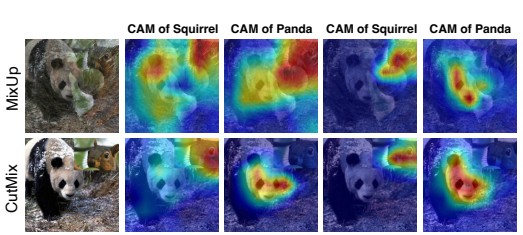

Figure 1:   visualization of hard mixed sample mining by class activation mapping (CAM) [49] of ResNet-50 on ImageNet. From left to right, CAM of top-2 predicted classes using mixup cross-entropy (MCE) and decoupled mixup (DM) loss.

---

*Equal contribution.   †Stan Z. Li (Stan.ZQ.Li@westlake.edu.cn) is the corresponding author.

37th Conference on Neural Information Processing Systems (NeurIPS 2023).

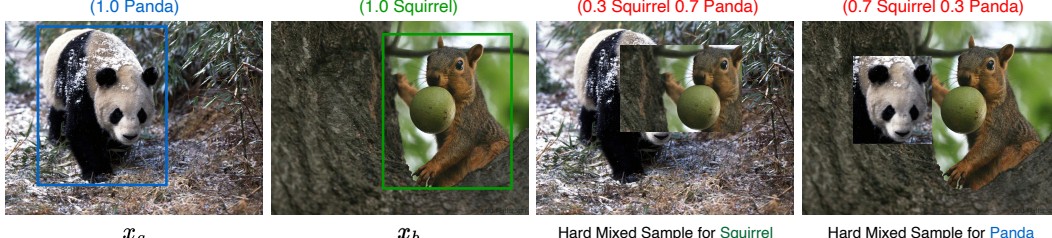

| (1.0 Panda) | (1.0 Squirrel) | (0.3 Squirrel 0.7 Panda) | (0.7 Squirrel 0.3 Panda) |

$x_a$ $\qquad\qquad\qquad\qquad$ $x_b$ $\qquad\qquad$ Hard Mixed Sample for Squirrel $\qquad$ Hard Mixed Sample for Panda

Figure 2: Illustration of the two types of hard mixed samples in CutMix with 'Squirrel' and 'Panda' as an example. Hard mixed samples indicate that the mixed sample contains salient features of a class, but the value of the corresponding label is small. MCE loss fails to leverage these samples.

end, regularization techniques have been deployed to improve generalization [61], which can be categorized into data-independent or data-dependent ones [16]. Some data-independent strategies, for example, constrain the model by punishing the parameters' norms, such as weight decay [40]. Among data-dependent strategies, data augmentations [51] are widely used. The augmentation policies often rely on particular domain knowledge [58] in different fields.

Mixup [77], a data-dependent augmentation technique, is proposed to generate virtual samples by a linear combination of data pairs and the corresponding labels with the mixing ratio $\lambda \in [0, 1]$. Recently, a line of optimizable mixup methods are proposed to improve mixing policies to generate object-aware virtual samples by optimizing discriminative regions in the data space to match the corresponding labels [56, 23, 22] (referred to as *dynamic* methods). However, although the *dynamic* approach brings some performance gain, the extra computational overhead degrades the efficiency of mixup augmentation significantly. Specifically, the most computation of *dynamic* methods is spent on optimizing label-mismatched samples, but the question of why these label-mismatched samples should be avoided during the mixup training has rarely been analyzed. In this paper, we find these mismatched samples are completely underutilized by *static* mixup methods, and the problem lies in the loss function, mixed cross-entropy loss (MCE). Therefore, we argue that these mismatched samples are not only not *static* mixup disadvantages but also hard mixed samples full of discriminative information. Taking CutMix [74] as an example, two types of hard mixed samples are shown on the *right* of Figure 2. Since MCE loss forces the model's predictions to be consistent with the soft label distribution, *i.e.,* the model cannot give high-confidence predictions for the relevant classes even if the feature is salient in hard mixed samples, we can say that these hard samples are not fully leveraged.

From this perspective, we expect the model to be able to mine these hard samples, *i.e.,* to give confident predictions according to salient features for localizing discriminative characteristics, even if the proportion of features is small. Motivated by this finding, we introduce simple yet effective Decoupled Mixup (DM) loss, a mixup objective function for explicitly leveraging the hard samples during the mixup training. Based on the standard mixed cross-entropy (MCE) loss, an extra decoupled regularizer term is introduced to enhance the ability to mine underlying discriminative statistics in the mixed sample by independently computing the predicted probabilities of each mixed class. Figure 1 shows the proposed DM loss can empower the *static* mixup methods to explore more discriminative features. Extensive experiments demonstrate that DM achieves data-efficiency training on supervised and semi-supervised learning benchmarks. Our contributions are summarized below:

- Unlike those dynamic mixup policies that design complicated mixing policies, we propose DM, a mixup objective function of mining discriminative features adaptively.

- Our work contributes more broadly to understanding mixup training: it is essential to focus not only on the smoothness by regression of the mixed labels but also on discrimination by encouraging the model to give reliable and confident predictions.

- Not only in supervised learning but the proposed DM can also be easily generalized to semi-supervised learning with a minor modification. By leveraging the unlabeled data, it can reduce the conformation bias and significantly improve performance.

- Comprehensive experiments on various tasks verify the effectiveness of DM, *e.g.*, DM-based *static* mixup policies achieve a comparable or even better performance than *dynamic* methods without the extra computation.

## 2 Related Work

**Mixup Augmentation.** As data-dependent augmentation techniques, mixup methods generate new samples by mixing samples and corresponding labels with well-designed mixing policies [77, 57, 69, 64]. The pioneering mixing method is Mixup [77], whose mixed samples are generated by linear interpolation between pairs of samples. ManifoldMix variants [57, 14] extend Mixup to the latent space of DNNs. After that, cut-based methods [74] are proposed to improve the mixup for localizing important features, especially in the vision field. Many researchers explore using nonlinear or optimizable sample mixup policies to generate more reliable mixed samples according to mixed labels, such as PuzzleMix variants [23, 22, 45], SaliencyMix variants [56, 60], AutoMix variants [38, 31], and SuperMix [11]. Concurrently, recent works try to generate more accurate mixed labels with saliency information [20] or attention maps [5, 9, 7] for Transformer architectures, which require prior pre-trained knowledge or attention information. On the contrary, the proposed decoupled mixup is a pluggable learning objective for mixup augmentations. Moreover, mixup methods extend to more than two elements [22, 11] and regression tasks [70]. Some researchers also utilize mixup augmentations to enhance contrastive learning [8, 21, 28, 50, 31] or masked image modeling [33, 6] to learn general representation in a self-supervised manner.

**Semi-supervised Learning and Transfer Learning.** Pseudo-Labeling [27] is a popular semi-supervised learning (SSL) method that utilizes artificial labels converted from teacher model predictions. MixMatch [2] and ReMixMatch [1] apply mixup on labeled and unlabeled data to enhance the diversity of the dataset. More accurate pseudo-labeling relies on data augmentation techniques to introduce consistency regularization, *e.g.*, UDA [65] and FixMatch [52] employ weak and strong augmentations to improve the consistency. Furthermore, CoMatch [29] unifies consistency regularization, entropy minimization, and graph-based contrastive learning to mitigate confirmation bias. Recently proposed works [62, 4] that improve FixMatch by designing more accurate confidence-based pseudo-label selection strategies, *e.g.*, FlexMatch [76] applying curriculum learning for updating confidence threshold dynamically and class-wisely. More recently, SemiReward [30] proposes a reward model to filter out accurate pseudo labels with reward scores. Fine-tuning a pre-trained model on labeled datasets is a widely adopted form of transfer learning (TL) in various applications. Previously, [13, 44] show that transferring pre-trained AlexNet features to downstream tasks outperforms hand-crafted features. Recent works mainly focus on better exploiting the discriminative knowledge of pre-trained models from different perspectives. L2-SP [35] promotes the similarity of the final solution with pre-trained weights by a simple L2 penalty. DELTA [34] constrains the model by a subset of pre-trained feature maps selected by channel-wise attention. BSS [68] avoids negative transfer by penalizing smaller singular values. More recently, Self-Tuning variants [67, 54] combined contrastive learning with TL to tackle confirmation bias and model shift issues in a one-stage framework.

## 3 Decoupled Mixup

### 3.1 Preliminary

**Mixed Cross-Entropy Underutilizes Mixup** Let us define $y \in \mathbb{R}^C$ as the ground-truth label with $C$ categories. For labeled data point $x \in \mathbb{R}^{\mathcal{W} \times \mathcal{H} \times \mathcal{C}}$ whose embedded representation $z$ is obtained from the model $M$ and the predicted probability $p$ can be calculated through a Softmax function $p = \sigma(z)$. Given the mixing ratio $\lambda \in [0, 1]$ and $\lambda$-related mixup mask $H \in \mathbb{R}^{\mathcal{W} \times \mathcal{H}}$, the mixed sample $(x_{(a,b)}, y_{(a,b)})$ can be generated as $x_{(a,b)} = H \odot x_a + (1 - H) \odot x_b$, and $y_{(a,b)} = \lambda y_a + (1 - \lambda)y_b$, where $\odot$ denotes element-wise product, $(x_a, y_a)$ and $(x_b, y_b)$ are sampled from a labeled dataset $L = \{(x_a, y_a)\}_{a=1}^{n_L}$. Note that superscripts denote the index; subscripts indicate the type of data, *e.g.*, $x_{(a,b)}$ represents a mixed sample generated from $x_a$ and $x_b$; $y^i$ indicates the label value on $i$-th position. Since the mixup labels are obtained by somehow $\lambda$-based interpolation, the standard CE loss weighted by $\lambda$, $\mathcal{L}_{CE} = y_{(a,b)}^T \log \sigma(z_{(a,b)})$, is typically used as the objective in the mixup training:

$$\mathcal{L}_{MCE} = -\sum_{i=1}^{C} \left( \lambda \mathbb{I}(y_a^i = 1) \log p_{(a,b)}^i + (1 - \lambda)\mathbb{I}(y_b^i = 1) \log p_{(a,b)}^i \right). \tag{1}$$

where $\mathbb{I}(\cdot) \in \{0, 1\}$ is an indicator function that values one if and only if the input condition holds. Noticeably, these two items of Equation 1 are classifying $y_a$ and $y_b$ while keeping the linear

consistency with mixing coefficient $\lambda$. As a result, DNNs with this mixup consistency prefer relatively less confident results in high-entropy behaviour [46] and longer training time in practice. **The main reason is that in addition to $\lambda$ constraint, the competing relationships defined by Softmax in $\mathcal{L}_{MCE}$ are the main cause of the confidence drop, which is more obvious when dealing with hard mixed samples.** Precisely, the competition between the mixed class $a$ and $b$ in Equation 1 can severely affect the prediction of a single class; that is, interference from other classes prevents the model from focusing its attention. This typically causes the model to be insensitive to the salient features of the target and thus undermines model performance, as shown in Figure 1. Although the *dynamic* mixup alleviates this problem, the extra time overhead is unavoidable if only focusing on mixing policies on the data level. Therefore, the key challenge is to design an ideal objective function for mixup training that maintains the smoothness of the mixup and can simultaneously explore the discriminative features without any computation costs.

## 3.2 Decoupled Regularizer

To achieve the above goal, we first dive into the $\mathcal{L}_{MCE}$ and propose the efficient decoupled mixup.

**Proposition 1.** *Assuming $x_{(a,b)}$ is generated from two different classes, minimizing $\mathcal{L}_{MCE}$ is equivalent to regress corresponding $\lambda$ in the gradient:*

$$
(\nabla_{z_{(a,b)}} \mathcal{L}_{MCE})^i = \begin{cases} -\lambda + \frac{\exp(z^i_{(a,b)})}{\sum_c \exp(z^c_{(a,b)})}, & i = a \\ -(1 - \lambda) + \frac{\exp(z^i_{(a,b)})}{\sum_c \exp(z^c_{(a,b)})}, & i = b \\ \frac{\exp(z^i_{(a,b)})}{\sum_c \exp(z^c_{(a,b)})}, & i \neq a, b \end{cases} \tag{2}
$$

**Softmax Degrades Confidence.** As we can see from Proposition 1, the predicted probability of $x_{(a,b)}$ will be consistent with $\lambda$, and the probability is computed from the Softmax directly. The Softmax forces the sum of predictions to one (winner takes all), which is undesirable in mixup classification, especially when there are multiple and non-salient targets in mixed samples, *e.g.,* hard mixed samples, as shown in Figure 2. The standard Softmax in $\mathcal{L}_{MCE}$ deliberately suppresses confidence and produces high-entropy predictions by coupling all classes. As a consequence, $\mathcal{L}_{MCE}$ makes many static mixup methods require longer epochs than vanilla training to achieve the desired results [57, 73]. Based on previous analysis, a novel mixup objective, decoupled mixup (DM), is raised to remove the Coupler and thus utilize the hard mixed samples adaptively, finally improving the performance of mixup methods. Specifically, for mixed data points $z_{(a,b)}$ generated from a random pair in labelled dataset $L$, an encoded mixed representation $z_{(a,b)} = f_\theta(x_{(a,b)})$ is generated by a feature extractor $f_\theta$. A mixed categorical probability of $i$-th class is attained:

$$
\sigma(z_{(a,b)})^i = \frac{\exp(z^i_{(a,b)})}{\sum_c \exp(z^c_{(a,b)})}. \tag{3}
$$

**Decoupled Softmax.** where $\sigma(\cdot)$ is standard Softmax. Equation 3 shows how the mixed probabilities are computed for a mixed sample. The competition between $a$ and $b$ is the main reason that results in low confidence of the model, *i.e.,* the sum of semantic information of hard mixed samples are larger than "1" defined by Softmax. Therefore, we propose to simply remove the competitor class in Equation 3 to achieve decoupled Softmax. The score on $i$-th class is not affected by the $j$-th class:

$$
\phi(z_{(a,b)})^{i,j} = \frac{\exp(z^i_{(a,b)})}{\cancel{\exp(z^j_{(a,b)})} + \sum_{c \neq j} \exp(z^c_{(a,b)})}. \tag{4}
$$

where $\phi(\cdot)$ is the proposed decoupled Softmax. In Equation 4, by removing the competitor, compared with Equation 1, the decoupled Softmax makes all items associated with $\lambda$ become -1 in gradient, the derivation is given in the A.1. Our Proposition 2 verifies that the expected results are achieved with decoupled Softmax.

**Proposition 2.** *With the decoupled Softmax defined above, decoupled mixup cross-entropy $\mathcal{L}_{DM}$ can boost the prediction confidence of the interested classes mutually and escape from the $\lambda$-constraint:*

$$
\mathcal{L}_{DM} = -\sum_{i=1}^{c} \sum_{j=1}^{c} y_a^i y_b^j \left( \log \big( \frac{p^i_{(a,b)}}{1 - p^j_{(a,b)}} \big) + \log \big( \frac{p^j_{(a,b)}}{1 - p^i_{(a,b)}} \big) \right). \tag{5}
$$

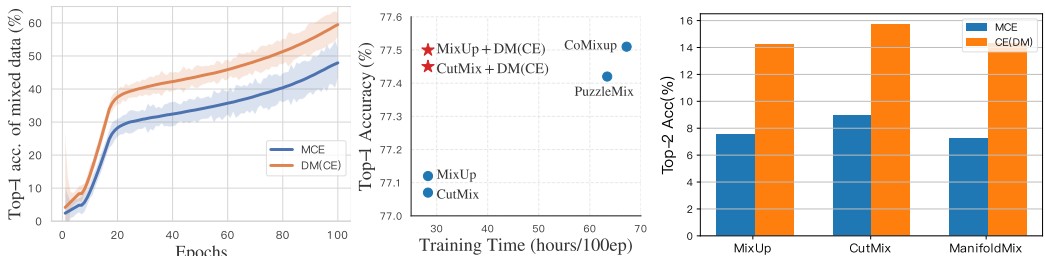

Figure 3: Results illustration of applying decouple mixup. *Left*: taking MixUp as an example, our proposed decoupled mixup cross-entropy, DM(CE), significantly improves training efficiency by exploring hard mixed sample; *Middle*: Acc *vs.* cost on ImageNet-1k; *Right*: Top-2 acc is calculated when the top-2 predictions equal to $\{y_a, y_b\}$.

**The Decoupled Mixup.** The proofs of Proposition 1 and 2 are given in the Appendix. In practice, the original smoothness of $\mathcal{L}_{MCE}$ should not be lost, and thus the proposed DM is a regularizer for discriminability. The final form of decoupled mixup can be formulated as follows:

$$\mathcal{L}_{DM(CE)} = -\big( \underbrace{y_{(a,b)}^T \log(\sigma(z_{(a,b)}))}_{\mathcal{L}_{MCE}} + \eta \underbrace{y_{[a,b]}^T \log(\phi(z_{(a,b)}))y_{[a,b]}}_{\mathcal{L}_{DM}} \big).$$

where $y_{(a,b)}$ indicates the mixed label while $y_{[a,b]}$ is two-hot label encoding, $\eta$ is a trade-off factor. Notice that $\eta$ is robust and can be set according to the character of mixup methods (see Sec. 5.4).

*Practical consequences of such simple modification on mixup and the performance:*

**Make What Should be Certain More Certain.** As we expected, mixup training with a decoupling mechanism will be more accurate and confident in handling hard mixed samples with our artificially constructed hard mixed samples by using PuzzleMix. Figure 3 *right* demonstrates the model trained with decoupled mixup mostly doubled the top-2 accuracy on these mixed samples, which also verifies the information contained in mixed samples is beyond the "1" defined by standard Softmax. More interestingly, this advantage of decoupled mixup, *i.e.,* higher confidence and accuracy, can be further amplified in semi-supervised learning due to the uncertainty of pseudo-labeling.

**Enhance the Training Efficiency.** It is straightforward to notice that there is no extra computation cost when using DM in vanilla mixup training, and the performance we can achieve is the same or even better than optimizable mixup policies, *i.e.,* PuzzleMix, CoMixup, *etc.* Figure 3 *left* and *middle* show decoupled mixup unveils the power of static mixup for more accurate and faster.

## 4 Extensions of Decoupled Mixup

With the high-accurate nature of decoupled mixup for mining hard mixed samples, semi-supervised learning is a suitable scenario to propagate the accurate label from labeled space to unlabeled space by using asymmetrical mixup. In addition, we can also generalize the decoupled mechanism into the binary cross-entropy for boosting the multi-classification task.

### 4.1 Asymmetrical Strategy for Semi-supervised Learning

Based on labeled data $L = \{(x_a, y_a)\}_{a=1}^{n_L}$, if we further consider unlabeled data $U = \{(u_a, v_a)\}_{a=1}^{n_U}$ decoupled mixup can be the strong connection between $L$ and $U$. Recall the confirmation bias [67] problem of SSL: the performance of the student model is restricted by the teacher model when learning from inaccurate pseudo-labels. To fully use the $L$ and strengthen the teacher model to provide more robust and accurate predictions, the unlabeled data with large $\lambda$ can be used to mix with the labeled data to form hard mixed samples. With these hard mixed samples, we can employ decoupled mixup into semi-supervised learning effectively. Since only the label of $L$ is accurate, we need to make a little asymmetric modification to the decoupled mixup, called Asymmetrical Strategy(AS). Formally, given the labeled and unlabeled datasets $L$ and $U$, AS builds reliable connection by generating hard mixed samples between $L$ and $U$ in an asymmetric manner ($\lambda < 0.5$):

$$\hat{x}_{(a,b)} = \lambda x_a + (1-\lambda)u_b; \quad \hat{y}_{(a,b)} = \lambda y_a + (1-\lambda)v_b.$$

Due to the uncertainty of the pseudo-label, only the labeled part is retained in $\mathcal{L}_{DM}$:

$$\hat{\mathcal{L}}_{DM} = y_a^T \log\left(\phi(z_{(a,b)})\right)y_b,$$

where $y_a$ and $y_b$ are one-hot labels from $L$. AS could be regarded as a special case of DM that only decouples on labeled data. Simply replacing $\mathcal{L}_{DM}$ with $\hat{\mathcal{L}}_{DM}$ can leverage the hard samples and alleviate the confirmation bias in semi-supervised learning.

## 4.2 Decoupled Binary Cross-entropy Loss

**Binary Cross-entropy Form of DM.** Different from Softmax-based classification, we can also build decoupled mixup in multi-label classification tasks (1-*vs*-all) by using mixup binary cross-entropy (MBCE) loss [63] ($\sigma(\cdot)$ denotes Sigmoid rather Softmax in this case). Proposition 2 demonstrates the decoupled CE can mutually enhance the confidence of predictions for the interested classes and be free from $\lambda$ limitations. Similarly, for MBCE, since it is not inherently bound to mutual interference between classes by Softmax, we have to preserve partial consistency and encourage more confident predictions, and thus propose a decoupled mixup binary cross-entropy loss, DM(BCE).

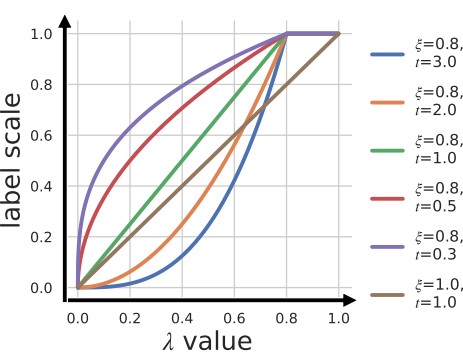

Figure 4: Rescaled label of different $\lambda$ value.

To this end, a rescaling function $r : \lambda, t, \xi \rightarrow \lambda'$ is designed to achieve this goal. The mixed label is rescaled by $r(\cdot)$: $y_{mix} = \lambda_a y_a + \lambda_b y_b$, where $\lambda_a$ and $\lambda_b$ are rescaled. The rescaling function is defined as follows:

$$r(\lambda, t, \xi) = \left(\frac{\lambda}{\xi}\right)^t, \quad 0 \le t, 0 \le \xi < 1, \tag{6}$$

where $\xi$ is the threshold, $t$ is an index to control the convexity. As shown in Figure 4, Equation 6 has three situations: (a) when $\xi = 0$, $t = 0$, the rescaled label is always equal to 1, as two-hot encoding; (b) when $\xi = 1$, $t = 1$, $r(\cdot)$ is a linear function (vanilla mixup); (c) the rest curves demonstrate $t$ is the parameter that changes the concavity and $\xi$ is responsible for truncating.

**Empirical Results.** In the case of interpolation-based mixup methods (*e.g.*, Mixup, ManifoldMix, *etc.*) that keep linearity between the mixed label and sample, the decoupled mechanism can be introduced by only adjusting threshold $t$. In the case of cutting-based mixing policies (*e.g.*, CutMix, *etc.*) where the mixed samples and labels have a square relationship (generally a convex function), we can approximate the convexity by adjusting $\xi$, which are detailed in Sec. 5.4 and Appendix C.5.

## 5 Experiments

We adopt two types of top-1 classification accuracy (Acc) metrics (the mean of three trials): (i) the median top-1 Acc of the last 10 epochs [52, 38] for supervised image classification tasks with Mixup variants, and (ii) the best top-1 Acc in all checkpoints for SSL tasks. Popular ConvNets and Transformer-based architectures are used as backbone networks: ResNet variants including ResNet [19] (R), Wide-ResNet (WRN) [75], and ResNeXt-32x4d (RX) [66], Vision Transformers including DeiT [55] and Swin Transformer (Swin) [37].

### 5.1 Image Classification Benchmarks

This subsection evaluates performance gains of DM on six image classification benchmarks, including CIFAR-100 [25], Tiny-ImageNet (Tiny) [10], ImageNet-1k [48], CUB-200-2011 (CUB) [59], FGVC-Aircraft (Aircraft) [42]. There are mainly two types of mixup methods based on their mixing policies: *static* methods including Mixup [77], CutMix [74], ManifoldMix [57], SaliencyMix [56], FMix [17], and ResizeMix [47], and *dynamic* mixup methods including PuzzleMix [23], AutoMix [38], and SAMix [31]. For a fair comparison, we use the optimal $\alpha$ in $\{0.1, 0.2, 0.5, 0.8, 1.0, 2.0\}$ for all mixup

Table 1: Top-1 Acc (%)↑ of small-scale image classification on CIFAR-100 and Tiny-ImageNet datasets based on ResNet variants.

| Datasets | CIFAR-100 | | | | | | Tiny-ImageNet | | | |
|---|---|---|---|---|---|---|---|---|---|---|
| | R-18 | | RX-50 | | WRN-28-8 | | R-18 | | RX-50 | |
| Methods | MCE | DM(CE) | MCE | DM(CE) | MCE | DM(CE) | MCE | DM(CE) | MCE | DM(CE) |
| Mixup | 79.12 | **80.44** | 82.10 | **82.96** | 82.82 | **83.51** | 63.86 | **65.07** | 66.36 | **67.70** |
| CutMix | 78.17 | **79.39** | 81.67 | **82.39** | 84.45 | **84.88** | 65.53 | **66.45** | 66.47 | **67.46** |
| ManifoldMix | 80.35 | **81.05** | 82.88 | **83.15** | 83.24 | **83.72** | 64.15 | **65.45** | 67.30 | **68.48** |
| FMix | 79.69 | **80.12** | 81.90 | **82.74** | 84.21 | **84.47** | 63.47 | **65.34** | 65.08 | **66.96** |
| ResizeMix | 80.01 | **80.26** | 81.82 | **82.96** | 84.87 | 84.72 | 63.74 | **64.33** | 65.87 | **68.56** |
| Avg. Gain | | +0.78 | | +0.77 | | +0.34 | | +1.18 | | +1.62 |

Table 2: Top-1 Acc (%)↑ of image classification on ImageNet-1k with ResNet variants using PyTorch-style 100-epoch training recipe.

| Methods | R-18 | | R-34 | | R-50 | |
|---|---|---|---|---|---|---|
| | MCE | DM(CE) | MCE | DM(CE) | MCE | DM(CE) |
| Vanilla | 70.04 | - | 73.85 | - | 76.83 | - |
| Mixup | 69.98 | **70.20** | 73.97 | **74.26** | 77.12 | **77.41** |
| CutMix | 68.95 | **69.26** | 73.58 | **73.88** | 77.07 | **77.32** |
| ManifoldMix | 69.98 | **70.33** | 73.98 | **74.25** | 77.01 | **77.30** |
| FMix | 69.96 | **70.26** | 74.08 | **74.34** | 77.19 | **77.38** |
| ResizeMix | 69.50 | **69.90** | 73.88 | **74.00** | 77.42 | **77.65** |
| Avg. Gain | | +0.32 | | +0.24 | | +0.25 |

Table 3: Top-1 Acc (%)↑ of image classification on ImageNet-1k based on ResNet-50 using RSB A3 100-epoch training recipe.

| Methods | MCE | DM(CE) | MBCE (one) | MBCE (two) | DM(BCE) (one) |
|---|---|---|---|---|---|
| RSB | 76.49 | **77.72** | 78.08 | 76.95 | **78.43** |
| Mixup | 76.01 | **76.69** | 77.66 | 77.42 | **78.28** |
| CutMix | 76.47 | **77.22** | 77.62 | 67.54 | **78.21** |
| ManifoldMix | 76.14 | **76.93** | 77.01 | 67.78 | **78.20** |
| FMix | 76.09 | **76.87** | 77.76 | 73.44 | **78.11** |
| ResizeMix | 76.90 | **77.21** | 77.85 | 77.30 | **78.32** |
| Avg. Gain | | +0.76 | | -4.38 | +0.47 |

Table 4: Top-1 Acc (%)↑ of classification on ImageNet-1k with ViTs.

| Methods | DeiT-S | | Swin-T | |
|---|---|---|---|---|
| | MCE | DM(CE) | MCE | DM(CE) |
| DeiT | 79.80 | **80.37** | 81.28 | **81.49** |
| Mixup | 79.65 | **80.04** | 80.71 | **80.97** |
| CutMix | 79.78 | **80.20** | 80.83 | **81.05** |
| FMix | 79.41 | **79.89** | 80.37 | **80.54** |
| ResizeMix | 79.93 | **80.03** | 80.94 | **81.01** |
| Avg. Gain | | +0.39 | | +0.19 |

Table 5: Top-1 Acc (%)↑ of fine-grained image classification on CUB-200 and FGVC-Aircrafts with ResNet variants.

| Datasets | CUB-200 | | | | FGVC-Aircrafts | | | |
|---|---|---|---|---|---|---|---|---|
| | R-18 | | RX-50 | | R-18 | | RX-50 | |
| Methods | MCE | DM(CE) | MCE | DM(CE) | MCE | DM(CE) | MCE | DM(CE) |
| Mixup | 78.39 | **79.90** | 84.58 | **85.04** | 79.52 | **82.66** | 85.18 | **86.68** |
| CutMix | 78.40 | **78.76** | 85.68 | **85.97** | 78.84 | **81.64** | 84.55 | **85.75** |
| ManifoldMix | 79.76 | **79.92** | 86.38 | **86.42** | 80.68 | **82.57** | 86.60 | **86.92** |
| FMix | 77.28 | **80.10** | 84.06 | **84.85** | 79.36 | **80.44** | 84.85 | **85.04** |
| ResizeMix | 78.50 | **79.58** | 84.77 | **84.92** | 78.10 | **79.54** | 84.08 | **84.51** |
| Avg. Gain | | +1.19 | | +0.35 | | +2.07 | | +0.73 |

algorithms and follow original hyper-parameters in papers. We adopt the open-source codebase OpenMixup [32] for most mixup methods. The detailed training recipes and hyper-parameters are provided in Appendix B. We also evaluate the adversarial robustness of CIFAR-100 in Appendix C.2.

**Small-scale Classification Benchmarks.** For small-scale classification benchmarks on CIFAR-100 and Tiny, we adopt the CIFAR version of ResNet variants and train with SGD optimizer following the common training settings [23, 38]. Table 1 and A2 show small-scale classification results. The proposed DM(CE) significantly improves MCE based on various mixup algorithms. Based on CIFAR-100 and three different CNNs (R-18, RX-50, and WRN-28-8), the decoupled mixup brings an average performance gain of 0.78%, 0.77%, and 0.34%, respectively. More notably, the gains on the Tiny dataset are significant, with average performance gains of: 1.18% and 1.62% on R-18 and RX-50.

**ImageNet and Fine-grained Classification Benchmarks.** For experiments on ImageNet-1k, we follow three popular training procedures: PyTorch-style setting [19], DeiT [55] setting, and RSB A3 [63] setting to demonstrate the generalizability of decoupled mixup. As shown in Table 2, 3, and 4, DM(CE) improves consistently over MCE in all mixup algorithms on three training settings we considered. The relative improvements have been calculated in the last row of tables. For example, DM(CE) yields around +0.4% for mixup methods based on ResNet variants using PyTorch-style and RSB A3 settings; around +0.5% and +0.2% for all methods based on DeiT-S and Swin-T using DeiT setting. Notice that MBCE(two) denotes using two-hot encoding for corresponding mixing classes, which yield worse performance than MBCE, and DM(BCE) adjusts the labels for the mixing classes by Equation 6. It verifies the necessity of DM(BCE) in the case of using MBCE. As for fine-grained benchmarks, we follow the training settings in AutoMix and initialize models with the official PyTorch pre-trained models (as supervised transfer learning). Table 5 and A6 show that DM(CE) noticeably boosts the original MCE for eight popular mixup variants, especially bringing 0.53%∼3.14% gains on Aircraft based on ResNet-18.

## 5.2 Semi-supervised Transfer Learning Benchmarks

Following the transfer learning (TL) benchmarks [71], we perform TL experiments on CUB, Aircraft, and Stanford-Cars [24] (Cars). Besides the vanilla Fine-Tuning baseline, we compare current

Table 6: Top-1 Acc (%)↑ of semi-supervised transfer learning on various TL benchmarks (CUB-200, FGVC-Aircraft, and Standfold-Cars) using only 15%, 30% and 50% labels based on ResNet-50.

| Methods | CUB-200 | | | FGVC-Aircraft | | | Stanford-Cars | | |
|---|---|---|---|---|---|---|---|---|---|
| | 15% | 30% | 50% | 15% | 30% | 50% | 15% | 30% | 50% |
| Fine-Tuning | $45.25_{\pm0.12}$ | $59.68_{\pm0.21}$ | $70.12_{\pm0.29}$ | $39.57_{\pm0.20}$ | $57.46_{\pm0.12}$ | $67.93_{\pm0.28}$ | $36.77_{\pm0.12}$ | $60.63_{\pm0.18}$ | $75.10_{\pm0.21}$ |
| **+DM** | $50.04_{\pm0.17}$ | $61.39_{\pm0.24}$ | $71.87_{\pm0.23}$ | $43.15_{\pm0.22}$ | $61.02_{\pm0.15}$ | $70.38_{\pm0.18}$ | $41.30_{\pm0.16}$ | $62.65_{\pm0.21}$ | $77.19_{\pm0.19}$ |
| BSS | $47.74_{\pm0.23}$ | $63.38_{\pm0.29}$ | $72.56_{\pm0.17}$ | $40.41_{\pm0.12}$ | $59.23_{\pm0.31}$ | $69.19_{\pm0.13}$ | $40.57_{\pm0.12}$ | $64.13_{\pm0.18}$ | $76.78_{\pm0.21}$ |
| Co-Tuning | $52.58_{\pm0.53}$ | $66.47_{\pm0.17}$ | $74.64_{\pm0.36}$ | $44.09_{\pm0.67}$ | $61.65_{\pm0.32}$ | $72.73_{\pm0.08}$ | $46.02_{\pm0.18}$ | $69.09_{\pm0.10}$ | $80.66_{\pm0.25}$ |
| **+DM** | $\mathbf{54.96}_{\pm0.65}$ | $\mathbf{68.25}_{\pm0.21}$ | $\mathbf{75.72}_{\pm0.37}$ | $\mathbf{49.27}_{\pm0.83}$ | $\mathbf{65.60}_{\pm0.41}$ | $\mathbf{74.89}_{\pm0.17}$ | $\mathbf{51.78}_{\pm0.34}$ | $\mathbf{74.15}_{\pm0.29}$ | $\mathbf{83.02}_{\pm0.26}$ |
| Self-Tuning | $64.17_{\pm0.47}$ | $75.13_{\pm0.35}$ | $80.22_{\pm0.36}$ | $64.11_{\pm0.32}$ | $76.03_{\pm0.25}$ | $81.22_{\pm0.29}$ | $72.50_{\pm0.45}$ | $83.58_{\pm0.28}$ | $88.11_{\pm0.29}$ |
| +Mixup | $62.38_{\pm0.32}$ | $74.65_{\pm0.24}$ | $81.46_{\pm0.27}$ | $59.38_{\pm0.31}$ | $74.65_{\pm0.26}$ | $81.46_{\pm0.27}$ | $70.31_{\pm0.27}$ | $83.63_{\pm0.23}$ | $88.66_{\pm0.21}$ |
| **+DM** | $\mathbf{73.06}_{\pm0.38}$ | $\mathbf{79.50}_{\pm0.35}$ | $\mathbf{82.64}_{\pm0.24}$ | $\mathbf{67.57}_{\pm0.27}$ | $\mathbf{80.71}_{\pm0.25}$ | $\mathbf{84.82}_{\pm0.26}$ | $\mathbf{81.69}_{\pm0.23}$ | $\mathbf{89.22}_{\pm0.21}$ | $\mathbf{91.26}_{\pm0.19}$ |
| Avg. Gain | **+5.95** | **+2.77** | **+1.34** | **+5.65** | **+4.52** | **+2.65** | **+7.22** | **+4.22** | **+2.35** |

Table 7: Top-1 Acc (%)↑ of semi-supervised learning on CIFAR-100 (using 400, 2500, and 10000 labels) based on WRN-28-8. Notice that DM denotes using DM(CE) and AS, Con denotes various unsupervised consistency losses, Rot denotes the rotation loss in ReMixMatch, and CPL denotes the curriculum labeling in FlexMatch.

| Methods | Losses | CIFAR-10 | | CIFAR-100 | | |
|---|---|---|---|---|---|---|
| | | 250 | 4000 | 400 | 2500 | 10000 |
| Pseudo-Labeling | CE | $53.51_{\pm2.20}$ | $84.92_{\pm0.19}$ | $12.55_{\pm0.85}$ | $42.26_{\pm0.28}$ | $63.45_{\pm0.24}$ |
| MixMatch | CE+Con | $86.37_{\pm0.59}$ | $93.34_{\pm0.26}$ | $32.41_{\pm0.66}$ | $60.24_{\pm0.48}$ | $72.22_{\pm0.29}$ |
| ReMixMatch | CE+Con+Rot | $93.70_{\pm0.05}$ | $95.16_{\pm0.01}$ | $57.15_{\pm1.05}$ | $73.87_{\pm0.35}$ | $79.08_{\pm0.27}$ |
| **MixMatch+DM** | CE+Con+DM | $89.16_{\pm0.71}$ | $95.15_{\pm0.68}$ | $35.72_{\pm0.53}$ | $62.51_{\pm0.37}$ | $74.70_{\pm0.28}$ |
| UDA | CE+Con | $94.84_{\pm0.06}$ | $95.71_{\pm0.07}$ | $53.61_{\pm1.59}$ | $72.27_{\pm0.21}$ | $77.51_{\pm0.23}$ |
| FixMatch | CE+Con | $95.14_{\pm0.09}$ | $95.79_{\pm0.08}$ | $53.58_{\pm0.82}$ | $71.97_{\pm0.16}$ | $77.80_{\pm0.12}$ |
| FlexMatch | CE+Con+CPL | $95.02_{\pm0.09}$ | $95.81_{\pm0.01}$ | $\mathbf{60.06}_{\pm1.62}$ | $73.51_{\pm0.20}$ | $78.10_{\pm0.15}$ |
| FixMatch+Mixup | CE+Con+MCE | $95.05_{\pm0.23}$ | $95.83_{\pm0.19}$ | $50.61_{\pm0.73}$ | $72.16_{\pm0.18}$ | $78.75_{\pm0.14}$ |
| **FixMatch+DM** | CE+Con+DM | $\mathbf{95.23}_{\pm0.09}$ | $\mathbf{95.87}_{\pm0.11}$ | $59.75_{\pm0.95}$ | $\mathbf{74.12}_{\pm0.23}$ | $\mathbf{79.58}_{\pm0.17}$ |
| Average Gain | | **+1.44** | **+0.95** | **+4.74** | **+2.30** | **+2.13** |

state-of-the-art TL methods, including BSS [68], Co-Tuning [71], and Self-Tuning [67]. For a fair comparison, we use the same hyper-parameters and augmentations as Self-Tuning, detailed in Appendix B.2. In Table 6, we adopt DM(CE) and AS for Fine-Tuning, Co-Tuning, and Self-Tuning using Mixup. DM(CE) and AS steadily improve Mixup and the baselines by large margins, *e.g.*, +4.62%~9.19% for 15% labels, +2.02%~5.67% for 30% labels, and +2.09%~3.15% for 50% labels on Cars. This outstanding improvement implies that generating mixed samples efficiently is essential for data-limited scenarios. A similar performance will be presented as well in the next SSL setting.

## 5.3 Semi-supervised Learning Benchmarks

Following [52, 76], we adopt the most commonly used CIFAR-10/100 datasets among the famous SSL benchmarks based on WRN-28-2 and WRN-28-8. We mainly evaluate the proposed DM on popular SSL methods MixMatch [2] and FixMatch [52], and compare with Pesudo-Labeling [27], ReMixMatch [1], UDA [65], and FlexMatch [76]. For a fair comparison, we use the same hyperparameters and training settings as the original papers and conduct experiments with the open-source codebase TorchSSL [76], detailed in Appendix B.2. Table 7 shows that adding DM(CE) and AS significantly improves MixMatch and FixMatch: DM(CE) brings 1.81~2.89% gains on CIFAR-10 and 1.27~3.31% gains on CIFAR-100 over MixMatch while bringing 1.78~4.17% gains on CIFAR-100 over FixMatch. Meanwhile, we find that directly applying mixup augmentations to FixMatch brings limited improvements, while FixMatch+DM achieves the best performance in most cases on CIFAR-10/100 datasets. Appendix C.3 provides further studies with limited labeled data. Therefore, mixup augmentations with DM can achieve data-efficient training in SSL.

## 5.4 Ablation Study and Analysis

**Hyperparameters and Proposed Components.** Since we have demonstrated the effectiveness of DM in the above experiments, Figure 1 and 5 verified that DM could well explore hard mixed samples. We then verify whether DM is robust to hyper-parameters (full hyper-parameters in Appendix C.5) and study the effectiveness of AS in SSL:

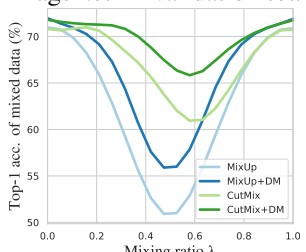

Figure 5: Top-1 Acc (%) of mixed samples on ImageNet-1k validation set.

Table 8: Ablation of the proposed asymmetric strategy (AS) and DM(CE) upon Self-Tuning for semi-supervised transfer learning on CUB-200 based on R-18.

| Methods | 15% | 30% | 50% | 100% |
|---|---|---|---|---|
| Self-Tuning | 57.82 | 69.12 | 73.59 | 75.08 |
| +MCE | 63.36 | 72.81 | 75.73 | 76.67 |
| +MCE+AS($\lambda \geq 0.5$) | 59.04 | 69.67 | 74.89 | 75.96 |
| +MCE+AS($\lambda \leq 0.5$) | 62.97 | 72.46 | 75.40 | 76.34 |
| **+DM(CE)+AS($\lambda \leq 0.5$)** | **66.17** | **74.25** | **77.68** | **78.52** |

(1) The only hyper-parameter $\eta$ in DM(CE) and DM(BCE) can be set according to the types of mixup methods. We grid search $\eta$ in $\{0.01, 0.1, 0.5, 1, 2\}$ on ImageNet-1k. As shown in Figure A2 *left*, the *static* (Mixup and CutMix) and the *dynamic* methods (PuzzleMix and AutoMix) prefer $\eta = 0.1$ and $\eta = 1$, respectively, which might be because the *dynamic* variants generate more discriminative and reliable mixed samples than the *static* methods.

(2) Hyper-parameters $\xi$ and $t$ in DM(BCE) can also be determined by the characters of mixup policies. We grid search $\xi \in \{1, 0.9, 0.8, 0.7\}$ and $t \in \{2, 1, 0.5, 0.3\}$. Figure A2 *middle* and *right* show that cutting-based methods (CutMix and AutoMix) prefer $\xi = 0.8$ and $t = 1$, while the interpolation-based policies (Mixup and ManifoldMix) use $\xi = 1.0$ and $t = 0.5$.

(3) Table 8 shows the superiority of AS($\lambda \leq 0.5$) in comparison to MCE and AS($\lambda \geq 0.5$), while using DM(CE) and AS($\lambda \leq 0.5$) further improves MCE.

(4) The experiments of different sizes of training data are performed to verify the data efficiency of DM. We can observe that decoupled mixup improves by around 2% accuracy without any computational overhead. The detailed results are shown in Appendix C.3.

**Occlusion Robustness** We also analyzed robustness against random occlusion [43] for models trained on ImageNet-1k using the official implementation[2]. Concretely, the classifier is thought to be robust if it predicts the correct label given an occluded version of the image. In other words, the network learns essential features (*e.g.,* semantic regions) that discriminate each class. For occlusion, we consider patch-based random masking. In particular, we split the image of $224 \times 224$ resolutions into patch size $16 \times 16$ and randomly mask $M$ patches out of the total number of $N$ patches, where the occlusion ratio is defined as $\frac{M}{N}$. As shown in Figure 6, the

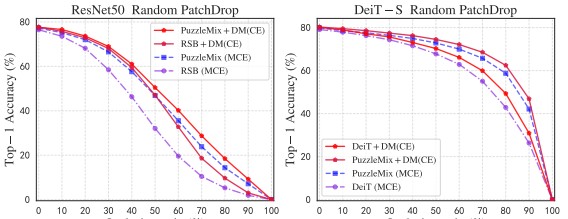

Figure 6: Robustness against different occlusion ratios of images for mixup augmentations using the MCE and our DM(CE) loss based on ResNet-50 (left) and DeiT-S (right) on ImageNet-1k. RSB and DeiT denote using CutMix+Mixup (*static* mixup policies) in RSB A3 [63] and DeiT [55] training settings. DM(CE) improves mixups by exploring hard mixed samples.

proposed DM helps various mixup methods achieve better occlusion robustness, indicating DM can force the model to learn discriminative features, *e.g.,* image patches with semantic information that is deterministic when the occlusion ratio is high.

## 6 Conclusion, Limitations, and Border Impacts

**Decoupled Mixup and Dynamic Mixups.** We investigate and show two limitations of the decoupled mixup. Different from *static* mixup methods, *dynamic* mixup spends extra time to optimize mixing masks in input space to align the mixed samples and labels. Although the optimized mixing policies can enhance the model to find discriminative features [38], their predictions are also under-confident.

Table 9: Top-1 Acc (%)↑ on CIFAR-100 and Tiny.

| Datasets | CIFAR-100 | | Tiny-ImageNet | |
|---|---|---|---|---|
| Backbone | WRN-28-8 | | RX-50 | |
| Methods | MCE | **DM(CE)** | MCE | **DM(CE)** |
| PuzzleMix | 85.02 | **85.25** | 67.83 | **68.04** |
| +RSB | 85.24 | **85.61** | 68.17 | **68.86** |
| AutoMix | 85.18 | **85.38** | 70.72 | **71.56** |
| +RSB | 85.35 | **85.54** | 70.98 | **72.37** |

In Table 9, we tried some advanced *dynamic* mixup policies, *e.g.,*

---

[2]https://github.com/Muzammal-Naseer/Intriguing-Properties-of-Vision-Transformers

PuzzleMix [23], and AutoMix [38], with decoupled mixup, and found the improvement is limited. The main reason is there will not be many hard mixed samples in *dynamic* mixups. Therefore, we additionally incorporate two *static* mixups in RSB training settings, *i.e.,* a half probability that Mixup or CutMix will be selected during the training. As expected, the improvements from the decoupled mixup are getting obvious upon *static* mixup variants. This is a very preliminary attempt that deserves more exploration in future works, and we provide more results of *dynamic* mixups in Appendix C.

Meanwhile, we further conduct comprehensive comparison experiments with modern Transformer-based architectures on CIFAR-100, considering the concurrent work TransMix [5] and TokenMix [36]. As shown in Table 10, where results with † denote the official implementation and the other are based on OpenMixup [32], DM(CE) enables DeiT (CutMix and Mixup) to achieve competitive performances as *dynamic* mixup variants like AutoMix and SAMix [31] based on ConvNeXt-S without introducing extra computational costs, while still performing worse than them based on DeiT-S. Compared with specially designed label mixing methods using attention maps, DM(CE) also achieves competi-

Table 10: Top-1 Acc (%)↑ of on CIFAR-100 training 200 and 600 epochs based on DeiT-S and ConvNeXt-T. Underlines denote the top-3 best results. Total training hours and GPU memory are collected on a single A100 GPU.

| Methods | DeiT-Small | | | | ConvNeXt-Tiny | | | |
|---|---|---|---|---|---|---|---|---|
| | 200 ep | 600 ep | Mem. | Time | 200 ep | 600 ep | Mem. | Time |
| Vanilla | 65.81 | 68.50 | 8.1 | 27 | 78.70 | 80.65 | 4.2 | 10 |
| Mixup | 69.98 | 76.35 | 8.2 | 27 | 81.13 | 83.08 | 4.2 | 10 |
| CutMix | 74.12 | 79.54 | 8.2 | 27 | 82.46 | 83.20 | 4.2 | 10 |
| DeiT | 75.92 | 79.38 | 8.2 | 27 | 83.09 | 84.12 | 4.2 | 10 |
| SmoothMix | 67.54 | 80.25 | 8.2 | 27 | 78.87 | 81.31 | 4.2 | 10 |
| SaliencyMix | 69.78 | 76.60 | 8.2 | 27 | 82.82 | 83.03 | 4.2 | 10 |
| AttentiveMix+ | 75.98 | 80.33 | 8.3 | 35 | 82.59 | 83.04 | 4.3 | 14 |
| FMix | 70.41 | 74.31 | 8.2 | 27 | 81.79 | 82.29 | 4.2 | 10 |
| GridMix | 68.86 | 74.96 | 8.2 | 27 | 79.53 | 79.66 | 4.2 | 10 |
| ResizeMix | 68.45 | 71.95 | 8.2 | 27 | 82.53 | 82.91 | 4.2 | 10 |
| PuzzleMix | 73.60 | 81.01 | 8.3 | 35 | 82.29 | 84.17 | 4.3 | 53 |
| AutoMix | 76.24 | 80.91 | 18.2 | 59 | 83.30 | 84.79 | 10.2 | 56 |
| SAMix | **77.94** | **82.49** | 21.3 | 58 | **83.56** | **84.98** | 10.3 | 57 |
| DeiT+TransMix | 76.17 | 79.33 | 8.4 | 28 | - | - | - | - |
| DeiT+TokenMix† | 76.25 | 79.57 | 8.4 | 34 | - | - | - | - |
| **DeiT+DM(CE)** | 76.20 | 79.92 | 8.2 | 27 | 83.44 | 84.49 | 4.2 | 10 |

tive performances to TransMix and TokenMix. How to further improve the decoupled mixup with the salient regions or dynamic attention information to research similar performances of *dynamic* mixing variants can also be studied in future works.

**The Next Mixup.** In a word, we introduce Decoupled Mixup (DM), a new objective function for considering both smoothness and mining discriminative features in mixup augmentations. The proposed DM helps *static* mixup methods (*e.g.,* MixUp and CutMix) achieve a comparable or better performance than the computationally expensive *dynamic* mixup policies. Most importantly, DM raises a question worthy of researching: *is it necessary to design very complex mixup policies?* We also find that decoupled mixup could be the bridge to combining *static* and *dynamic* mixup. However, the introduction of additional hyperparameters may take users some extra time to check on other than images or other mixup methods. This also leads to the core question of the next step in the development of this work: how to design a more elegant and adaptive mixup training objective that connects different types of mixups to achieve high data efficiency? We believe these explorations and questions can inspire future research in the community of mixup augmentations.

## Acknowledgement

This work was supported by National Key R&D Program of China (No. 2022ZD0115100), National Natural Science Foundation of China Project (No. U21A20427), and Project (No. WU2022A009) from the Center of Synthetic Biology and Integrated Bioengineering of Westlake University. We thank the AI Station of Westlake University for the support of GPUs and thank all reviewers for polishing the manuscript.

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

# Appendix

In the Appendix sections, we provide proofs of proposition 1 (§A.1) and proposition 2 (§A.2), implementation details (§B), and more results of comparison experiment and empirical analysis (§C).

## A  Proof of Proposition

### A.1  Proof of Proposition 1

**Proposition 1.** Assuming $x_{(a,b)}$ is generated from two different classes, minimizing $\mathcal{L}_{MCE}$ is equivalent to regress corresponding $\lambda$ in the gradient of $\mathcal{L}_{MCE}$:

$$(\nabla_{z_{(a,b)}}\mathcal{L}_{MCE})^l = \begin{cases} -\lambda + \frac{\exp(z^i_{(a,b)})}{\sum_c \exp(z^c_{(a,b)})}, & l = i \\ -(1-\lambda) + \frac{\exp(z^j_{(a,b)})}{\sum_c \exp(z^c_{(a,b)})}, & l = j \\ \frac{\exp(z^l_{(a,b)})}{\sum_c \exp(z^c_{(a,b)})}, & l \neq i, j \end{cases} \tag{7}$$

*Proof.* For the mixed sample $(x_{(a,b)}, y_{(a,b)})$, $z_{(a,b)}$ is derived from a feature extractor $f_\theta$ (i.e $z_{(a,b)} = f_\theta(x_{(a,b)})$). According to the definition of the mixup cross-entropy loss $\mathcal{L}_{MCE}$, we have:

$$\begin{aligned}
\left(\nabla_{z_{(a,b)}}\mathcal{L}_{MCE}\right)^l &= \frac{\partial \mathcal{L}_{MCE}}{\partial z^l_{(a,b)}} = -\frac{\partial}{\partial z^l_{(a,b)}}\left(y^T_{(a,b)}\log\left(\sigma(z_{(a,b)})\right)\right) \\
&= -\sum_{i=1}^C \left(y^i_{(a,b)}\frac{\partial}{\partial z^l_{(a,b)}}\left(\log\left(\frac{\exp(z^i_{(a,b)})}{\sum_{j=1}^C \exp(z^j_{(a,b)})}\right)\right)\right) \\
&= -\sum_{i=1}^C \left(y^i_{(a,b)}\frac{\sum_{j=1}^C \exp(z^j_{(a,b)})}{\exp(z^i_{(a,b)})}\frac{\partial}{\partial z^l_{(a,b)}}\left(\frac{\exp(z^i_{(a,b)})}{\sum_{j=1}^C \exp(z^j_{(a,b)})}\right)\right) \\
&= -\sum_{i=1}^C \left(y^i_{(a,b)}\left(\delta^l_i - \frac{\exp(z^l_{(a,b)})}{\sum_{j=1}^C \exp(z^j_{(a,b)})}\right)\right) \\
&= \frac{\exp(z^l_{(a,b)})}{\sum_{j=1}^C \exp(z^j_{(a,b)})} - y^l_{(a,b)}.
\end{aligned}$$

Similarly, we have:

$$\begin{aligned}
\left(\nabla_{z_{(a,b)}}\mathcal{L}_{DM}\right)^l &= \frac{\partial \mathcal{L}_{DM}}{\partial z^l_{(a,b)}} = -\frac{\partial}{\partial z^l_{(a,b)}}\left(y^T_{[a,b]}\log\left(H(z_{(a,b)})\right)y_{[a,b]}\right) \\
&= -\frac{\partial}{\partial z^l_{(a,b)}}\left(\sum_{i,j=1}^C y^i_a \log\left(\frac{\exp(z^i_{(a,b)})}{\sum_{k\neq j}^C \exp(z^j_{(a,b)})}\right)y^j_b + \sum_{i,j=1}^C y^j_a \log\left(\frac{\exp(z^i_{(a,b)})}{\sum_{k\neq i}^C \exp(z^j_{(a,b)})}\right)y^i_b\right) \\
&= -\sum_{i,j=1}^C \left(y^i_a y^j_b \frac{\partial}{\partial z^l_{(a,b)}}\left(\log\left(\frac{\exp(z^i_{(a,b)})}{\sum_{k\neq j}^C \exp(z^k_{(a,b)})}\right) + \log\left(\frac{\exp(z^j_{(a,b)})}{\sum_{k\neq i}^C \exp(z^k_{(a,b)})}\right)\right)\right) \\
&= -\sum_{i,j=1}^C \left(y^i_a y^j_b \left(\delta^l_i - \frac{\sum_{k\neq j}\exp(z^k_{(a,b)})\delta^l_k}{\sum_{k\neq j}\exp(z^k_{(a,b)})} + \delta^l_j - \frac{\sum_{k\neq i}\exp(z^k_{(a,b)})\delta^l_k}{\sum_{k\neq i}\exp(z^k_{(a,b)})}\right)\right) \\
&= \frac{\sum_{k\neq i}\exp(z^k_{(a,b)})\delta^l_k}{\sum_{k\neq i}\exp(z^k_{(a,b)})} + \frac{\sum_{k\neq j}\exp(z^k_{(a,b)})\delta^l_k}{\sum_{k\neq j}\exp(z^k_{(a,b)})} - \delta^l_i - \delta^l_j.
\end{aligned}$$

Thus, for $\mathcal{L}_{DM}$ loss:

$$(\nabla_{z_{(a,b)}} \mathcal{L}_{MCE})^l = \begin{cases} -1 + \frac{\exp(z_{(a,b)}^i)}{\sum_{c \neq j} \exp(z_{(a,b)}^c)}, & l = i \\ -1 + \frac{\exp(z_{(a,b)}^j)}{\sum_{c \neq i} \exp(z_{(a,b)}^c)}, & l = j \\ \frac{\exp(z_{(a,b)}^l)}{\sum_{c \neq i} \exp(z_{(a,b)}^c)} + \frac{\exp(z_{(a,b)}^l)}{\sum_{c \neq j} \exp(z_{(a,b)}^c)}, & l \neq i, j \end{cases} \tag{8}$$

## A.2 Proof of Proposition 2

**Proposition 2.** With the decoupled Softmax defined above, decoupled mixup cross-entropy $\mathcal{L}_{DM(CE)}$ can boost the prediction confidence of the interested classes mutually and escape from the $\lambda$-constraint:

$$\mathcal{L}_{DM(CE)} = \sum_{i=1}^{c} \sum_{j=1}^{c} y_a^i y_b^j \Big( \log \big( \frac{p_{(a,b)}^i}{1 - p_{(a,b)}^j} \big) + \log \big( \frac{p_{(a,b)}^j}{1 - p_{(a,b)}^i} \big) \Big).$$

*Proof.* For the mixed sample $(x_{(a,b)}, y_{(a,b)})$, $z_{(a,b)}$ is derived from a feature extractor $f_\theta$ (i.e $z_{(a,b)=f_\theta(x_{(a,b)})}$). According to the definition of the mixup cross-entropy loss $\mathcal{L}_{DM(CE)}$, we have:

$$
\begin{aligned}
\mathcal{L}_{DM(CE)} &= y_{[a,b]}^T \log \big( H(Z_{(a,b)}) \big) y_{[a,b]} \\
&\triangleq y_a^T \log \big( H(Z_{(a,b)}) \big) y_b + y_b^T \log \big( H(Z_{(a,b)}) \big) y_a \\
&= \sum_{i,j=1}^{C} y_a^i \log \big( \frac{\exp(z_{(a,b)}^i)}{\sum_{k \neq j}^C \exp(z_{(a,b)}^j)} \big) y_b^j + \sum_{i,j=1}^{C} y_a^j \log \big( \frac{\exp(z_{(a,b)}^i)}{\sum_{k \neq i}^C \exp(z_{(a,b)}^j)} \big) y_b^i \\
&= \sum_{i,j=1}^{C} y_a^i y_b^j \big( \log \big( \frac{\exp(z_{(a,b)}^i)}{\sum_{k \neq j}^C \exp(z_{(a,b)}^j)} \big) + \log \big( \frac{\exp(z_{(a,b)}^j)}{\sum_{k \neq i}^C \exp(z_{(a,b)}^i)} \big) \big) \\
&= \sum_{i,j=1}^{C} y_a^i y_b^j \big( \log \big( \frac{\frac{\exp(z_{(a,b)}^i)}{\sum_{k=1}^C \exp(z_{(a,b)}^k)}}{\frac{\sum_{k \neq j}^C \exp(z_{(a,b)}^j)}{\sum_{k=1}^C \exp(z_{(a,b)}^k)}} \big) + \log \big( \frac{\frac{\exp(z_{(a,b)}^j)}{\sum_{k=1}^C \exp(z_{(a,b)}^k)}}{\frac{\sum_{k \neq i}^C \exp(z_{(a,b)}^k)}{\sum_{k=1}^C \exp(z_{(a,b)}^k)}} \big) \big) \\
&= \sum_{i,j=1}^{C} y_a^i y_b^j \big( \log \big( \frac{p_{(a,b)}^i}{1 - p_{(a,b)}^j} \big) + \log \big( \frac{p_{(a,b)}^j}{1 - p_{(a,b)}^i} \big) \big),
\end{aligned}
$$

where $p_{(a,b)} = \sigma(z_{(a,b)})$. $\qquad\qquad\square$

# B  Implementation Details

## B.1  Dataset

We briefly introduce used image datasets. (1) Small scale classification benchmarks: CIFAR-10/100 [25] contains 50,000 training images and 10,000 test images in $32 \times 32$ resolutions, with 10 and 100 classes settings. Tiny-ImageNet [10] is a rescaled version of ImageNet-1k, which has 10,000 training images and 10,000 validation images of 200 classes in $64 \times 64$ resolutions. (2) Large scale classification benchmarks: ImageNet-1k [26] contrains 1,281,167 training images and 50,000 validation images of 1000 classes in $224 \times 224$ resolutions. (3) Small-scale fine-grained classification scenarios: CUB-200-2011 [59] contains 11,788 images from 200 wild bird species for fine-grained classification. FGVC-Aircraft [42] contains 10,000 images of 100 classes of aircraft. Standford-Cars [24].

## B.2  Training Settings

**Small-scale image classification.** As for small-scale classification benchmarks on CIFAR-100 and Tiny-ImageNet datasets, we adopt the CIFAR version of ResNet variants, *i.e.*, using a $3 \times 3$ convolution instead of the $7 \times 7$ convolution and MaxPooling in the stem, and follow the common training settings [23, 38]: the basic data augmentation includes `RandomFlip` and `RandomCrop` with 4 pixels padding; SGD optimizer and Cosine learning rate Scheduler [39] are used with the SGD weight decay of 0.0001, the momentum of 0.9, and the Batch size of 100; all methods train 800 epochs with the basic learning rate $lr = 0.1$ on CIFAR-100 and 400 epochs with $lr = 0.2$ on Tiny-ImageNet.

**Fine-grained image classification.** As for fine-grained classification experiments on CUB-200 and Aircraft datasets, all mixup methods are trained 200 epochs by SGD optimizer with the initial learning rate $lr = 0.001$, the weight decay of 0.0005, and the batch size of 16. We use the standard augmentations `RandomFlip` and `RandomResizedCrop`, and load the official PyTorch pre-trained models on ImageNet-1k as initialization.

**ImageNet image classification.** For large-scale classification tasks on ImageNet-1k, we evaluate mixup methods on three popular training procedures, and Tab. A1 shows the full training settings of the three settings. Notice that DeiT [55] and RSB A3 [63] settings employ Mixup and CutMix with a switching probability of 0.5 during training. (a) PyTorch-style setting. Without any advanced training strategies, a PyTorch-style setting is used to study the performance gains of mixup methods: SGD optimizer is used to train 100 epochs with the SGD weight decay of 0.0001, a momentum of 0.9, a batch size of 256, and the basic learning rate of 0.1 adjusted by Cosine Scheduler. Notice that we replace the step learning rate decay with Cosine Scheduler [39] for better performances following [74]. (b) DeiT [55] setting. We use the DeiT setting to verify the DM(CE) effectiveness in training Transformer-based networks: AdamW optimizer [41] is used to train 300 epochs with a batch size of 1024, the basic learning rate of 0.001, and the weight decay of 0.05. (c) RSB A3 [63] setting. This setting adopts similar training techniques as DeiT to ConvNets, *especially using MBCE instead of MCE*: LAMB optimizer [72] is used to train 100 epochs with the batch size of 2048, the basic learning rate of 0.008, and the weight decay of 0.02. Notice that DeiT and RSB A3 settings use the combination of Mixup and CutMix (50% random switching probabilities) as the baseline.

**Semi-supervised transfer learning.** For semi-supervised transfer learning benchmarks, we use the same hyper-parameters and augmentations as Self-Tuning[3]: all methods are initialized by PyTorch pre-trained models on ImageNet-1k and trained 27k steps in total by SGD optimizer with the basic learning rate of 0.001, the momentum of 0.9, and the weight decay of 0.0005. We reproduced Self-Tuning and conducted all experiments in OpenMixup [32].

**Semi-supervised learning.** For semi-supervised learning benchmarks (training from scratch), we adopt the most commonly used CIFAR-10/100 datasets among the famous SSL benchmarks based on WRN-28-2 and WRN-28-8 following [52, 76]. For a fair comparison, we use the same hyperparameters and training settings as the original papers and adopt the open-source codebase TorchSSL [76] for all methods. Concretely, we use an SGD optimizer with a basic learning rate of

---

[3]https://github.com/thuml/Self-Tuning

Table A1: Ingredients and hyper-parameters used for ImageNet-1k training settings.

| Procedure | PyTorch | DeiT | RSB A3 |
|---|---|---|---|
| Train Res | $224^2$ | $224^2$ | $224^2$ |
| Test Res | $224^2$ | $224^2$ | $224^2$ |
| Test crop ratio | 0.875 | 0.875 | 0.95 |
| Epochs | 100/300 | 300 | 100 |
| Batch size | 256 | 1024 | 2048 |
| Optimizer | SGD | AdamW | LAMB |
| LR | 0.1 | $1 \times 10^{-3}$ | $8 \times 10^{-3}$ |
| LR decay | cosine | cosine | cosine |
| Weight decay | $10^{-4}$ | 0.05 | 0.02 |
| optimizer momentum | 0.9 | $\beta_1, \beta_2 = 0.9, 0.999$ | ✗ |
| Warmup epochs | ✗ | 5 | 5 |
| Label smoothing $\epsilon$ | ✗ | 0.1 | ✗ |
| Dropout | ✗ | ✗ | ✗ |
| Stoch. Depth | ✗ | 0.1 | 0.05 |
| Repeated Aug | ✗ | ✓ | ✓ |
| Gradient Clip. | ✗ | 1.0 | ✗ |
| H. flip | ✓ | ✓ | ✓ |
| RRC | ✓ | ✓ | ✓ |
| Rand Augment | ✗ | 9/0.5 | 6/0.5 |
| Auto Augment | ✗ | ✗ | ✗ |
| Mixup alpha | ✗ | 0.8 | 0.1 |
| Cutmix alpha | ✗ | 1.0 | 1.0 |
| Erasing prob. | ✗ | 0.25 | ✗ |
| ColorJitter | ✗ | ✗ | ✗ |
| EMA | ✗ | 0.99996 | ✗ |
| CE loss | ✓ | ✓ | ✗ |
| BCE loss | ✗ | ✗ | ✓ |

$lr = 0.03$ adjusted by Cosine Scheduler, the total $2^{20}$ steps, the batch size of 64 for labeled data, and the confidence threshold $\tau = 0.95$.

## B.3 Hyper-parameter Settings

We follow the basic hyper-parameter settings (*e.g.,* $\alpha$) for mixup variants in OpenMixup [32], where we reproduce most comparison methods. Notice that *static* methods denote Mixup [77], CutMix [74], ManifoldMix [57], SaliencyMix [56], FMix [17], ResizeMix [47], and *dynamic* methods denote PuzzleMix [23], AutoMix [38], and SAMix [31]). Similarly, *interpolation-based* methods denote Mixup and ManifoldMix while *cutting-based* methods denote the rest mixup variants mentioned above. We set the hyper-parameters of DM(CE) as follows: For CIFAR-100 and ImageNet-1k, *static* methods use $\eta = 0.1$, and *dynamic* methods use $\eta = 1$. For Tiny-ImageNet and fine-grained datasets, *static* methods use $\eta = 1$ based on ResNet-18 while $\eta = 0.1$ based on ResNeXt-50; *dynamic* methods use $\eta = 1$. As for the hyper-parameters of DM(BCE) on ImageNet-1k, *cutting-based* methods use $t = 1$ and $\xi = 0.8$, while *interpolation-based* methods use $t = 0.5$ and $\xi = 1$. Note that we use $\alpha = 0.2$ and $\alpha = 2$ for the *static* and *dynamic* methods when using the proposed DM.

Table A2: Top-1 Acc (%)↑ of small-scale image classification on CIFAR-100 and Tiny-ImageNet datasets based on ResNet variants.

| Datasets | CIFAR-100 | | | | | | Tiny-ImageNet | | | |
|---|---|---|---|---|---|---|---|---|---|---|
| | R-18 | | RX-50 | | WRN-28-8 | | R-18 | | RX-50 | |
| Methods | MCE | DM(CE) | MCE | DM(CE) | MCE | DM(CE) | MCE | DM(CE) | MCE | DM(CE) |
| SaliencyMix | 79.12 | **79.28** | 81.53 | **82.61** | 84.35 | **84.41** | 64.60 | **66.56** | 66.55 | **67.52** |
| PuzzleMix | 81.13 | **81.34** | 82.85 | **82.97** | 85.02 | **85.25** | 65.81 | **66.52** | 67.83 | **68.04** |
| AutoMix | 82.04 | **82.32** | 83.64 | **83.94** | 85.18 | **85.38** | 67.33 | **68.18** | 70.72 | **71.56** |
| SAMix | 82.30 | **82.40** | 84.42 | **84.53** | 85.50 | **85.59** | 68.89 | **69.16** | 72.18 | **72.39** |
| Avg. Gain | | **+0.19** | | **+0.40** | | **+0.15** | | **+0.95** | | **+0.56** |

Table A3: Top-1 Acc (%)↑ of image classification on ImageNet-1k with ResNet variants using PyTorch-style 100-epoch training recipe.

| Methods | R-18 | | R-34 | | R-50 | |
|---|---|---|---|---|---|---|
| | MCE | DM(CE) | MCE | DM(CE) | MCE | DM(CE) |
| SaliencyMix | 69.16 | **69.57** | 73.56 | **73.92** | 77.14 | **77.42** |
| PuzzleMix | 70.12 | **70.32** | 74.26 | **74.51** | 77.54 | **77.71** |
| AutoMix | 70.51 | **70.64** | 74.52 | **74.77** | 77.91 | **78.15** |
| SAMix | 70.85 | **70.90** | 74.96 | **75.10** | 78.11 | **78.36** |
| Avg. Gain | | **+0.20** | | **+0.25** | | **+0.23** |

Table A4: Top-1 Acc (%)↑ of image classification on ImageNet-1k based on ResNet-50 using RSB A3 100-epoch training recipe.

| Methods | MCE | DM(CE) | MBCE (one) | MBCE (two) | DM(BCE) (one) |
|---|---|---|---|---|---|
| SaliencyMix | 76.85 | **77.25** | 77.93 | 72.74 | **78.24** |
| PuzzleMix | 77.27 | **77.60** | 78.02 | 77.19 | **78.15** |
| AutoMix | 77.45 | **77.82** | 78.33 | 77.46 | **78.62** |
| SAMix | 78.33 | **78.45** | 78.64 | 77.58 | **78.75** |
| Avg. Gain | | **+0.30** | | -1.99 | **+0.04** |

Table A5: Top-1 Acc (%)↑ of classification on ImageNet-1k with ViTs.

| Methods | DeiT-S | | Swin-T | |
|---|---|---|---|---|
| | MCE | DM(CE) | MCE | DM(CE) |
| DeiT | 79.80 | **80.37** | 81.28 | **81.49** |
| SaliencyMix | 79.32 | **79.86** | 80.68 | **80.83** |
| PuzzleMix | 79.84 | **80.25** | 81.03 | **81.16** |
| AutoMix | 80.78 | **80.91** | 81.80 | **81.92** |
| SAMix | 80.94 | **81.12** | 81.87 | **81.97** |
| Avg. Gain | | **+0.32** | | **+0.13** |

Table A6: Top-1 Acc (%)↑ of fine-grained image classification on CUB-200 and FGVC-Aircrafts with ResNet variants.

| Datasets | CUB-200 | | | | FGVC-Aircrafts | | | |
|---|---|---|---|---|---|---|---|---|
| | R-18 | | RX-50 | | R-18 | | RX-50 | |
| Methods | MCE | DM(CE) | MCE | DM(CE) | MCE | DM(CE) | MCE | DM(CE) |
| SaliencyMix | 77.95 | **78.28** | 83.29 | **84.51** | 80.02 | **81.31** | 84.31 | **85.07** |
| PuzzleMix | 78.63 | **78.74** | 84.51 | **84.67** | 80.76 | **80.89** | 86.23 | **86.36** |
| AutoMix | 79.87 | **81.08** | 86.56 | **86.74** | 81.37 | **82.18** | 86.69 | **86.82** |
| SAMix | 81.11 | **81.27** | 86.83 | **86.95** | 82.15 | **83.68** | 86.80 | **87.22** |
| Avg. Gain | | **+0.45** | | **+0.42** | | **+0.94** | | **+0.36** |

# C  More Experiment Results

## C.1  Image Classification Benchmarks

**Small-scale classification benchmarks.** For small-scale classification benchmarks on CIFAR-100 and Tiny-ImageNet, we also conduct experiments of applying the proposed DM(CE) to *dynamic* mixup methods even though these algorithms have achieved high performance in Table A2: DM(CE) brings 0.23%∼0.36% on CIFAR-100 for the previous state-of-the-art PuzzleMix and brings 0.21%∼0.27% on Tiny-ImageNet for the current state-of-the-art method SAMix. Overall, the proposed DM(CE) produces +0.15∼0.4% and 0.56∼0.95% average gains on CIFAR-100 and Tiny-ImageNet, demonstrating its generalizability to advanced mixup augmentations.

**ImageNet and fine-grained classification benchmarks.** For experiments on ImageNet-1k, we also employ the proposed DM(CE) to *dynamic* mixup approaches on ImageNet-1k with PyTorch-style [19], DeiT [55], and RSB A3 [63] training settings to further evaluate the generalizability of decoupled mixup. As shown in Table A3 and Table A4, DM(CE) gains +0.2∼0.3% top-1 accuracy over MCE in average for four *dynamic* mixup methods based on ResNet variants on ImageNet-1k; Table A5 show DM(CE) also improves *dynamic* methods based on popular DeiT-S and Swin-T backbones with modern training recipes. These results indicate that the proposed decoupled mixup can also boost these *dynamic* mixup augmentations with high performances on ImageNet-1k. Moreover, the proposed DM(CE) can improve *dynamic* mixup variants on fine-grained classification benchmarks, as shown in Table A6, with around +0.4∼0.9% average gains over MCE based on ResNet variants.

Table A7: Top-1 Acc (%)↑ and FGSM error (%)↓ on CIFAR-100 and Tiny-ImageNet based on ResNet-18 training 400 epochs.

| Datasets | CIFAR-100 | | | | Tiny-ImageNet | | | |
|---|---|---|---|---|---|---|---|---|
| | Acc(%)↑ | | Error(%)↓ | | Acc(%)↑ | | Error(%)↓ | |
| Methods | MCE | DM(CE) | MCE | DM(CE) | MCE | DM(CE) | MCE | DM(CE) |
| Mixup | 79.34 | **79.70** | 70.28 | **70.05** | 63.86 | **65.07** | 89.06 | **88.91** |
| CutMix | 79.58 | **79.77** | 87.43 | **86.84** | 65.53 | **66.45** | 89.14 | **88.79** |
| ManifoldMix | 80.18 | **81.06** | 72.50 | **72.19** | 64.15 | **65.45** | 88.78 | **88.52** |
| PuzzleMix | 80.22 | **80.58** | 79.76 | **79.53** | 65.81 | **66.13** | **91.83** | 92.05 |
| AutoMix* | 81.78 | **81.96** | 69.94 | **69.80** | 67.33 | **68.18** | 88.37 | **88.34** |

## C.2  Adversarial Robustness

Since mixup variants are proven to enhance the robustness of DNNs against adversarial samples [77], we compare the robustness of the original MCE and the proposed DM(CE) by performing the

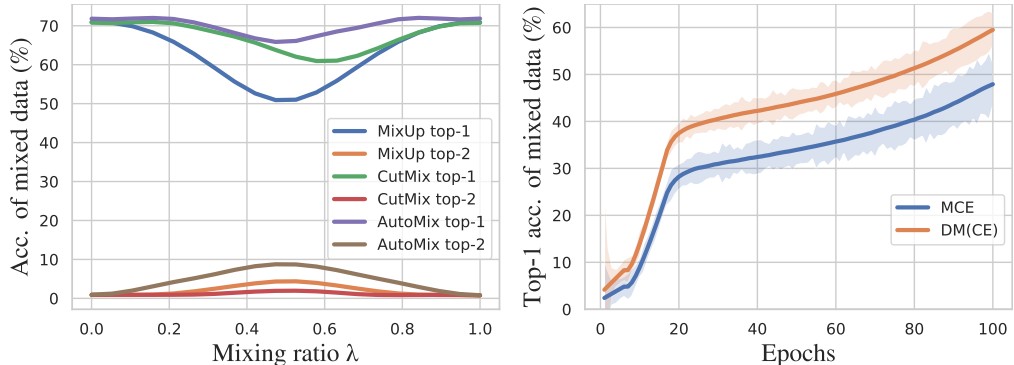

Figure A1: Experimental overviews of hard mixed sample mining. **Left**: Top-1 and top-2 accuracy of mixed data based on ResNet-50 trained 100 epochs on ImageNet-1k. Prediction is counted as correct if the top-1 prediction belongs to $\{y_a, y_b\}$; prediction is counted as correct if the top-2 predictions are equal to $\{y_a, y_b\}$. Compared with *static* policies like Mixup [77] and CutMix [74], the *dynamic* method AutoMix [38] significantly reduces the difficulty of mixup classification and alleviates the label mismatch issue [23] by providing more reliable mixed samples but also requires a large computational overhead. **Right**: Taking Mixup as an example, our proposed decoupled mixup cross-entropy, DM(CE), significantly improves training efficiency by exploring hard mixed samples and alleviates the label mismatch issue.

FGSM [15] white-box attack of 8/255 $\ell_\infty$ epsilon ball following [23]. Table A7 shows that DM(CE) improves top-1 Acc of MCE while maintaining the competitive FGSM error rates for five popular mixup algorithms, which indicates that DM(CE) can *boost discrimination without disturbing the smoothness properties* of mixup variants.

## C.3 Data-efficient Mixup with Limited Training Labels

To further DM whether data-efficient mixup training can be truly achieved, we conducted supervised experiments on CIFAR-100 with different sizes of training data. 15%, 30%, and 50% of the CIFAR-100 data are randomly selected as training data, and the test data are unchanged. The proposed decoupled mixup uses DM(CE) as the loss function by default. From Table A8, we can see that DM improves performance consistently without any computational overhead. Especially when using only 15% of the data, DM can improve accuracy by 2%. Therefore, combined with the experimental results of semi-supervised learning in Sec. 5.3 and Sec. 5.2, we can say that mixup training with DM is more data-efficient with limited data.

Table A8: Top-1 Acc (%)↑ of image classification on CIFAR-100 with ResNet-18 using 15%, 30%, and 50% labeled training sets.

| Methods | 15% | | 30% | | 50% | |
| --- | --- | --- | --- | --- | --- | --- |
| | MCE | DM(CE) | MCE | DM(CE) | MCE | DM(CE) |
| Vanilla | 42.48 | - | 56.41 | - | 64.32 | - |
| Mixup | 42.23 | **44.39** | 55.61 | **56.78** | 64.55 | **65.92** |
| CutMix | 43.81 | **44.85** | 55.99 | **57.14** | 64.38 | **65.87** |
| SaliencyMix | 42.95 | **44.01** | 55.42 | **56.51** | 64.56 | **66.10** |
| PuzzleMix | 42.67 | **43.87** | 56.19 | **57.36** | 64.74 | **66.26** |
| Avg. Gain | | **+1.36** | | **+1.14** | | **+1.48** |

## C.4 Empirical Analysis

In addition to occlusion robustness in Figure 6, we analyze the top-1 and top-2 mixup classification accuracy and visualize validation accuracy curves during training to empirically demonstrate the effectiveness of DM in Figure A1.

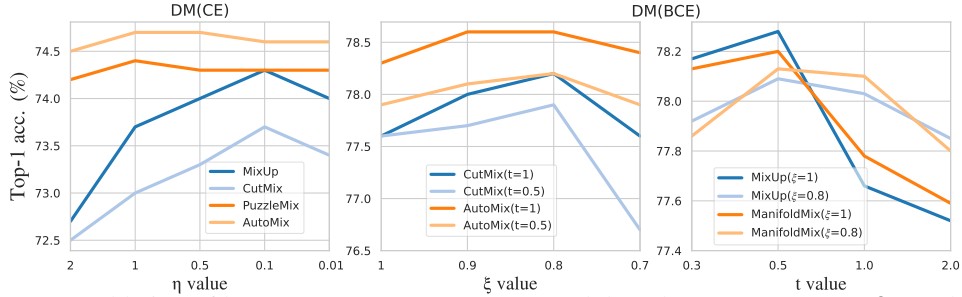

Figure A2: Ablation of hyper-parameters on ImageNet-1k based on ResNet-34. **Left**: analyzing the balancing weight $\eta$ in DM(CE); **Middle**: analyzing $\xi$ in DM(BCE) when $t$ is fixed to 1 and $0.5$; **Right**: analyzing $t$ in DM(BCE) when $\xi$ is fixed to 1 and $0.8$.

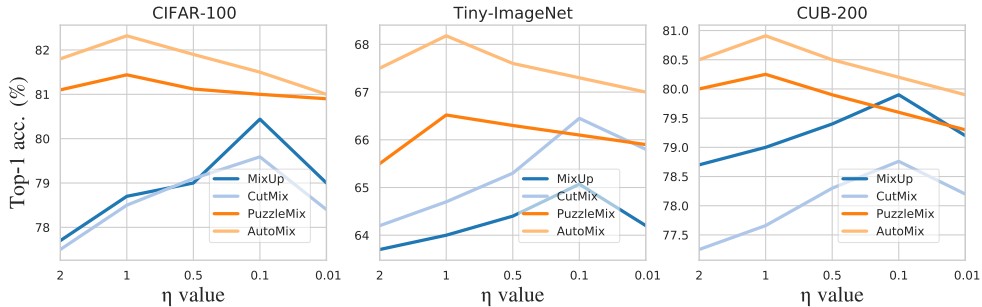

Figure A3: Sensitivity analysis of hyper-parameters on different datasets based on ResNet-18.

## C.5 Ablation Study and Analysis

**Ablation of hyper-parameters** We first provide ablation experiments of the shared hyper-parameter $\eta$ in DM(CE) and DM(BCE). In Figure A2 *left*, the *static* (Mixup and CutMix) and the *dynamic* methods (PuzzleMix and AutoMix) prefer $\eta = 0.1$ and $\eta = 1$, respectively, which might be because the *dynamic* variants generate more discriminative and reliable mixed samples than the *static* methods. Then, Figure A2 *middle* and *right* show that ablation studies of hyper-parameters $\xi$ and $t$ in DM(BCE), where cutting-based methods (CutMix and AutoMix) prefer $\xi = 0.8$ and $t = 1$, while the interpolation-based policies (Mixup and ManifoldMix) use $\xi = 1.0$ and $t = 0.5$.

**Sensitivity Analysis** To verify the robustness of hyper-parameter $\eta$, extra experiments are conducted on CIFAR-100, Tiny-ImageNet, and CUB-200 datasets. Figure A3 shows the results consistent with our ablation study in Sec. 5.4. *Dynamic* mixup methods prefer the large value of $\eta$ (*e.g.,* 1.0), while *static* ones are more like a small value (*e.g.,* 0.1). The main reason for this is the *dynamic* methods generate mixed samples where label mismatch is relatively rare, relying on larger weights to achieve better results, while the opposite is true in *static* methods.

