# OpenReview forum: "Harnessing Hard Mixed Samples with Decoupled Regularizer"
_NeurIPS.cc/2023/Conference — NeurIPS 2023 poster_

### Official Review · Reviewer_s9eU · 2023-06-29

**Soundness:** 3 good
**Presentation:** 3 good
**Contribution:** 3 good
**Rating:** 6
**Confidence:** 4

**Summary:**

This paper proposes Decoupled Softmax(Eq. 4), which is an interesting improvement to the previous mixup method, which mitigates the impact of noise in mixed samples by modifying the loss.

**Strengths:**

The proposed idea is simple and effective. The manuscript has a high degree of completion and is rich in experiments.

**Weaknesses:**

I see no obvious disadvantages.

There are some related literatures that I think are close to the author's claim:
The Benefits of Mixup for Feature Learning, It found that modifying the lambda of y in the mixup loss does not significantly affect the performance of the model.
UMIX: Improving Importance Weighting for Subpopulation Shift via Uncertainty-Aware Mixup, This paper also found that appropriate modifications can be made in the mixup loss to improve the generalization of the model.


**Questions:**

I have no other issues and the paper is written clearly and easily understood.

---

> ### Author Rebuttal · Authors · 2023-08-03
>
> Thank you for recognizing our work! These two papers you have mentioned are very interesting works, but many of the differences are worth discussing.
>
> - Although The Benefits of Mixup for Feature Learning argues the different linear interpolation parameters for features and labels can still achieve similar performance, their analysis is still limited to the standard Softmax, ignoring the fact that the semantic information of the mixed samples should be greater than "1". If the setting for the sum of the weights is greater than 1 and directional (not a completely random linear interpolation), then I suspect the conclusion is likely to change.
> - UMIX proposes a new mixup loss function from an uncertainty perspective, but his need for additional training time and hyperparameters to obtain the importance weight makes the method not so convenient.
>
> We will discuss them in the related works in the revised version. Please feel free to ask any other questions!

---

### Official Review · Reviewer_sDcS · 2023-07-05

**Soundness:** 2 fair
**Presentation:** 2 fair
**Contribution:** 3 good
**Rating:** 6
**Confidence:** 4

**Summary:**

The authors propose a new objective function with decoupled regularizer named decoupled mixup (DM) to harness hard mixed samples and mine discriminative features adaptively. This method is available on supervised learning and semi-supervised learning. Unlike the previous approaches which propose a more complicated dynamic mixup policy with extra computation, the proposed DM can adaptively utilize those hard mixed samples to mine discriminative features without losing the original smoothness of mixup.

**Strengths:**

1.	Harnessing hard mixed samples without losing the original smoothness of mixup is an interesting idea.
2.     The proposed DM enables static mixup methods to achieve comparable or even exceed the performance of dynamic methods without any extra computation.
3.	The authors provide lots of experiments to demonstrate the effectiveness of the proposed method.

**Weaknesses:**

1.	It is confusing about the important notations, such as i，j, a, b. The author should follow some notational conventions.
2.	Section 4.2 is abrupt with a poor description, confusing notation definitions, and low contextual relevance.
3.	Equation (2) is not closely related to the context. Please explain it in detail.
4.	Since Table 6 reports the experiments on transfer learning, the authors should describe that their proposed DM can be adapted to transfer learning in the related work. In addition, the title of Section 5.2 should be changed to "Transfer Learning Benchmarks". It might be more reasonable if the authors exchange Sections 5.2 and 5.3 since the authors pay more attention on semi-supervised learning.
5.	It would be better if the authors move lines 164 - 175 and Figure 3 to Section 5. Figures 5 and 6 are too small in the font and the colors of the lines are hard to distinguish.
6.	Some important equations are not numbered, and there should be a ',' link between the equation and the "where" in the same sentence.

**Questions:**

Please refer to the weaknesses.

**Limitations:**

The authors described potential negative societal impact.

---

> ### Author Rebuttal · Authors · 2023-08-03
>
> Thank you for your precious time and great efforts. Your insightful suggestions and professional questions are the key to improving the quality of the paper. We will address your questions one by one and make the corresponding changes in the reversion.
>
> ---
> ### Answers to questions
>
> > 1. It is confusing about the important notations, such as i，j, a, b. The author should follow some notational conventions.
>
> Thanks for your constructive suggestion for improving the readability, we will make the notation more clear in our revised version. Actually, we have followed the famous mixup method CutMix to define the notations of mixed samples from different two samples $x_a, x_b$ and use $i,j$ as indices to access vectors/matrices, which are conventional notations in most machine learning papers. To be more clear, taking the label $y$ as example, one-hot label $y_a\in\mathbb{R}^C$, mixed label $y_{(a,b)}\in\mathbb{R}^C$. Introduce $i,j$ to access specific values , $y^i_a\in\mathbb{R}$. Specifically,
> $$
> y_a^k=\\left\\{
> \\begin{aligned}
> 1& &&k=i\\\\
> 0& &&k\neq i
> \\end{aligned}\right.
> $$
> $$
> y_{(a,b)}^k=\\left\\{
> \\begin{aligned}
> \lambda &&k=i\\\\
> 1-\lambda &&k=j\\\\
> 0 && k\neq i,j
> \\end{aligned}\right.
> $$
>
> > 2. Section 4.2 is abrupt with a poor description, confusing notation definitions, and low contextual relevance.
> - Section 4.2 is an extension of DM to multi-label classification. Since the manuscript has been too compact, some of the detailed statements have been overlooked resulting in poor presentation in this section. To improve the presentation, in an updated version we will simplify section 3.2 (e.g., merge Equation 3 into 3.1 and reduce the length). Then in section 4.2, we will add back-to-back statements to improve the coherence, e.g., "softmax-based models cannot deal with multi-label problems, thus how to introduce DM mechanism in multi-label classification is a question worth considering.” To make the notation more clear, the rescaled $\lambda$ is harmonized as $\lambda'$ then in Line 212, the equation becomes $y_{(a,b)}=\lambda'_a y_a+\lambda'_by_b$.
>
> > 3. Equation (2) is not closely related to the context. Please explain it in detail.
> - Equation (2) shows minimizing $L_{MCE}$ is equivalent to a regression task taking $\lambda$ as labels during the optimization. Therefore, this equation is direct evidence that $L_{MCE}$ suppresses prediction confidence. This is the motivation why we need to propose DM. We will explain this more clear at the beginning of section 3.2 in our updated version.
>
> > 4. Since Table 6 reports the experiments on transfer learning, the authors should describe that their proposed DM can be adapted to transfer learning in the related work. In addition, the title of Section 5.2 should be changed to "Transfer Learning Benchmarks". It might be more reasonable if the authors exchange Sections 5.2 and 5.3 since the authors pay more attention on semi-supervised learning.
> - Thanks for your constructive again! We will adopt your suggestions in the revision: a) enriching the related works about transfer learning. b) change the title of section 5.2.
>
> > 5. It would be better if the authors move lines 164 - 175 and Figure 3 to Section 5. Figures 5 and 6 are too small in the font and the colors of the lines are hard to distinguish.
> - Sure, we will re-organize the paper according to your valuable suggestions.
>
> > Some important equations are not numbered, and there should be a ',' link between the equation and the "where" in the same sentence.
> - Thanks for this careful review, we will fix them in the revision.

---

> > ### Comment · Reviewer_sDcS · 2023-08-22
> > **Response to Rebuttal**
> >
> > Thank you for your response and clarifying many details to the questions. I increase the score from 5 to 6.

---

### Official Review · Reviewer_y8Fo · 2023-07-07

**Soundness:** 3 good
**Presentation:** 3 good
**Contribution:** 3 good
**Rating:** 7
**Confidence:** 3

**Summary:**

This paper introduced a simple strategy decoupled mixup (DM) to improve the effectiveness of Mixup and its variants. Regarding the softmax result of a mixed image with a pair of classes, one class is removed from the denominator when computing the loss of the other class.  Authors provided both theoretical and empirical analyses to confirm DM has the effect of increasing the confidence of the predicted classes. DM can also be applied in semi-supervised learning and multi-label classification. Extensive experiments have been conducted covering standard image classification, semi-supervised learning and semi-supervised fine-tuning.

**Strengths:**

Strengths
1. The work is well motivated with the idea of making confident predictions for Mixup training.
2. The proposed idea is novel and can be combined with existing Mixup variants.
3. DM is proved to be effective on various tasks.


**Weaknesses:**

1. Authors claimed the smoothness of Mixup can be preserved by DM, but I didn't see detailed discussions about this. The idea of DM seems contradict with smoothness of Mixup or label smoothing. To this end, the mechanism of DM is not totally clear.
2. Why PuzzleMix and AutoMix are not evaluated in image classification tasks?


**Questions:**

See Weaknesses

---

> ### Author Rebuttal · Authors · 2023-08-03
>
> Thank you for your great efforts, these valuable questions and constructive suggestions, which are exactly what the paper needs. We will take your suggestions and solve your problems one by one, and all corresponding changes will be reflected in the revision.
>
> ---
>
> ### Answers to questions
>
> >1.  Authors claimed the smoothness of Mixup can be preserved by DM, but I didn't see detailed discussions about this. The idea of DM seems contradict with smoothness of Mixup or label smoothing. To this end, the mechanism of DM is not totally clear.
>
> - The final form of $L_{DM(CE)}$ is composed of two parts: $L_{MCE}$ and $L_{DM}$. Our claim in Line 160 is that $L_{DM}$ is the role of the regularizer in mining hard mixed samples to improve the discriminability of the model. According to Equation 2, we can clearly see that the optimization of $L_{MCE}$ can be regarded as a regression task with $\lambda$ as the target, which brings smoothness to decision boundaries. Therefore, we say that decoupled mixup (DM) has both the properties of smoothness and enhanced discrimination at the same time. In a word, The mechanism of DM is that when dealing with mixed samples with information greater than "1" (Line 135-139), it can break the limit of $L_{MCE}$ to fully utilize the extra information from hard mixed samples.
>
> > 2. Why PuzzleMix and AutoMix are not evaluated in image classification tasks?
>
> - Due to the efficiency of data augmentation, static methods are still mainly used in the main text. However, according to L331 **" View results of dynamic mixups in the Appendix."**, in our supplemental submission, we have experimented with not only AutoMix and PuzzleMix in full accordance with the setup of the main text, but also included other dynamic methods, SaliencyMix and SAMix, as detailed in Tables A2, A3, A4, A5, A6, A7, and A8.

---

### Official Review · Reviewer_JzuZ · 2023-07-12

**Soundness:** 4 excellent
**Presentation:** 3 good
**Contribution:** 3 good
**Rating:** 8
**Confidence:** 3

**Summary:**

The authors point out that while $\textit{dynamic}$ mixup methods are shown to be effective, they induce too much computational cost. To address this issue, they propose a $\textit{static}$ method called Decoupled Mixup, which utilizes the hard mixed samples. The authors suggest that the Softmax function will suppress the model's confidence on hard mixed samples. Based on this idea, the authors propose a decoupled mixup cross-entropy loss which uses a decoupled version of Softmax that ease the "sum-to-one" constraint of Softmax. This loss is then added in the standard mixup loss as a regularization term.

Empirically, the authors show that decoupled mixup improves the top-1 accuracy performance beyond some standard mixup methods on a variety of benchmark datasets. They also show that this method can be generalized with good performance on semi-supervised learning.

**Strengths:**

1. The idea of utilizing hard mixed samples from the perspective of Softmax function is novel and interesting.

2. The theoretical explanation of the effectiveness of the proposed algorithm is solid.

3. The experiments are conducted thoroughly on plenty of tasks, datasets and Mixup methods.

4. Experiments settings are explained in details, especially the configurations of the hyperparameters of the proposed new method DM, leaving great convenience for future practitioners trying to reproduce the work.

5. The improvement of DM beyond standard Mixup methods is empirically shown to be significant. Also, while there isn't much improvement of DM beyond dynamic Mixup, the saved computational costs are also valuable.



**Weaknesses:**

1. The typesettings of some figures and tables are a bit too dense.

2. A few typo.

**Questions:**

1. Line 72. Should the word "conformation" be "confirmation"?

2. Proposition 1, line 133. What does it mean by saying "$\textit{to regress corresponding}\ \lambda$" in the gradient Equation (2)?

3. Line 144, "... for mixed data point $z_{(a,b)}$". Should it be "$x_{(a,b)}$"?

4. Line 147, "$\textbf{Decoupled Softmax}$". The subtitle seems not in the right place. Probably layout error.

5. Line 144-147. From my understanding the text here is to provide the definition of Softmax function and to introduce $\sigma(\cdot)$ as its denotation. I would simply put it in Section 3.1 before Equation (1), along with the descriptions of all other notations.

6. Line 148-150. The text here is basically telling the same story as line 137-140, that Softmax suppresses the confidence of the model on hard mixed samples, the sum of whose semantic information should be more than $1$. I think they can be combined together, rather than having the idea repeated twice in one paragraph.

7. Equation (4), "$\phi(z_{(a,b)})^{i,j}$". Since it is suggested in the previous text that superscripts denote the index, I think here $j$ can be put as a subscript of $\phi$ as an indication of the function, making the expression clearer.

8. Line 153, "the decoupled Softmax makes all items associated with $\lambda$ becomes $-1$ in gradient". Though the proof of this statement should be straightforward, can you provide it in the Appendix as well, since it's mentioned that "the derivatiopn is given in the A.1" while there is only a proof of Equation 1 in A.1?

9. Line 180, "multi-classification task". Should it be "multi-lable classification task" to be more specific?

10. Line 186, "... the unlabeled data with large $\lambda$ ..." . Does it mean unlabeled data with large combination weight? i.e. the $(1-\lambda)$ as in "$\hat{x}_{(a,b)}=\lambda{x_a}+(1-\lambda)u_b$" actually?

11. Equation under line 192. The expression in RHS (particularly $z_{(a,b)}$) is not clear enough to indicate that only the labeled part is retained in $\mathcal{L}_{DM}$.

12. Line 220 "threshold $t$" and line 222 "$\xi$". The notations are used exchanged.

13. Appendix A.1, first line of the equations under line 526. Should the minus sign "$-$" here be the equal sign "$=$"?

14. The idea of raising confidence for hard mixed samples is interesting, but conventionally from the perspective of calibration, one may wish the confidence to not be too large. Will there be a contradict (or a trade-off) between leveraging hard mixed samples and improving the models' calibration performance?

15. In some tasks (datasets), mixup may not necessarily create hard mixed samples, especially when the type of data doesn't have apparent semantic information, for example points on a 2D plane. Also, when manifold intrusion occurs, the true label of a mixed sample may differs from both the labels of the pair of real samples. In these cases, DM may provide no significant improvement, or even degrade the performance. Do you have any insight or principle to determine the "level of necessity" or "effectiveness" of applying DM in a given task and dataset?

**Limitations:**

Not many obvious limitations. The derivation of some theory statement is not complete.

---

> ### Author Rebuttal · Authors · 2023-08-03
>
> Thanks for your great effort and very constructive comments to help us improve the manuscript. We will address your questions one by one and make the corresponding changes in the reversion. Please note that due to compilation issues, $L$ denotes $\mathcal{L}$
>
> ---
> ### Answers to questions
> > 1, 3, 4, 5, 6, 7, 9, 12, and 13, typos and writing suggestions.
>
> - Thanks so much for your careful feedback! We'll correct all typos and adopt your suggestions to polish the writing of this paper in the revised version.
>
> > 2. Proposition 1, line 133. What does it mean by saying "to regress corresponding $\lambda$" in the gradient Equation (2)?
>
> - Here “regress to corresponding $\lambda$” means equation (2) can be regarded as a regression task with the $\lambda$ as a target when $i=a$ or $b$. Because there are gradients if the predicted probability is not exactly equal to $\lambda$. Maybe proposition 1 could be more clear if it becomes “Assuming $x_{(a,b)}$ is generated from two different classes, minimizing $L_{MCE}$ is equivalent to a regression task taking $\lambda$ as labels”. Besides, this equation is an explanation that implies $L_{MCE}$ suppresses prediction confidence.
>
> > 8. Line 153, "the decoupled Softmax makes all items associated with $\lambda$ becomes -1 in gradient". Can you provide it in the Appendix as well?
> - Sure, similar with A.1 we have:
>
> $$
> \\begin{align*}
>         \big( \nabla_{z(a, b)} L_{DM} \big)^{l}
>         &=\frac{\partial L_{DM}}{\partial z_{\tiny(a, b)}^l} = -\frac{\partial}{\partial z_{\tiny(a, b)}^l} \Big(y_{[a,b])}^{T}\log\big(H(z_{(a, b)})\big) y_{[a,b]} \Big) \\\\
>         &=-\frac{\partial}{\partial z_{\tiny(a, b)}^l} \Big( \sum_{i,j=1}^{C} y_a^i \log(\frac{\exp(z_{\tiny(a,b)}^i)}{\sum_{k \neq j}^{C}\exp(z_{\tiny(a,b)}^j)}) y_b^j+\sum_{i,j=1}^{C} y_a^j \log(\frac{\exp(z_{\tiny(a,b)}^i)}{\sum_{k \neq i}^{C}\exp(z_{\tiny(a,b)}^j)}) y_b^i \Big) \\\\
>         &=-\sum_{i,j=1}^{C}\Big( y_{a}^{i}y_{b}^j \frac{\partial}{\partial z_{\tiny(a, b)}^l}\big(\log(\frac{\exp(z_{\tiny(a,b)}^i)}{\sum_{k \neq j}^{C}\exp(z_{\tiny(a,b)}^k)}) + \log(\frac{\exp(z_{\tiny(a,b)}^j)}{\sum_{k \neq i}^{C}\exp(z_{\tiny(a,b)}^k)}) \big) \Big) \\\\
>         &=-\sum_{i,j=1}^{C}\Big( y_{a}^{i}y_{b}^j \big(\delta_i^l - \frac{\sum_{k \neq j}\exp(z_{\tiny(a,b)}^k)\delta_k^l}{\sum_{k \neq j}\exp(z_{\tiny(a,b)}^k)} + \delta_j^l - \frac{\sum_{k \neq i}\exp(z_{\tiny(a,b)}^k) \delta_k^l}{\sum_{k \neq i}\exp(z_{\tiny(a,b)}^k)} \big) \Big) \\\\
>         &=\frac{\sum_{k \neq i}\exp(z_{\tiny(a,b)}^k)\delta_k^l}{\sum_{k \neq i} \exp(z^k_{(a,b)})}+\frac{\sum_{k \neq j}\exp(z_{\tiny(a,b)}^k)\delta_k^l}{\sum_{k \neq j} \exp(z^k_{(a,b)})} - \delta_i^l - \delta_j^l.
> \\end{align*}
> $$
> Thus, for $L_{DM}$ loss:
>
> $$
> \\begin{align}(\nabla_{z_{(a,b)}} L_{MCE})^l=
>     \\begin{cases}
>         -1+\frac{\exp(z^i_{(a,b)})}{\sum_{c \neq j} \exp(z^c_{(a,b)})}, & l=i
>         \\\\
>         -1+\frac{\exp(z^j_{(a,b)})}{\sum_{c \neq i} \exp(z^c_{(a,b)})}, & l=j
>         \\\\
>         \frac{\exp(z^l_{(a,b)})}{\sum_{c \neq i} \exp(z^c_{(a,b)})}+\frac{\exp(z^l_{(a,b)})}{\sum_{c \neq j} \exp(z^c_{(a,b)})}, & l \neq i, j
>     \\end{cases}
> \\end{align}
> $$
>
> > 10. Line 186, does it mean unlabeled data with large combination weight?
> - Yes, it is. As we stated in Line 191, we set $\lambda<0.5$ to achieve this result. I'm sorry for confusing your understanding, the description here is not rigorous, we would change “large $\lambda$” to “large combination weight” in our revised version.
>
> > 11.  Equation under line 192. The expression is not clear to indicate that only the labeled part is retained in $L_{DM}$.
> - Given $ z_{(a,b)}$, the (pseudo) label index of $x_a,u_b$ is $i,j$, we have:
>
> $$
> \hat{L_{DM}}=log(\frac{exp(z^i_{(a,b)})}{\sum_{k\neq j}^C exp(z^j_{(a,b)})})
> $$
> This means calculating the prediction of class $i$ by decoupling class $j$. Written in matrix form this becomes $y_a^T log(\phi(z_{(a,b)}))y_b$, which means accessing the results of ground-truth label $y_a$ after coupling off the pseudo-label $y_b$. Thus, we say this formula retains the labeled part. To make it more clear, we will add an intuitive example in the revised version.
>
> > 14. Will there be a contradict (or a trade-off) between leveraging hard mixed samples and improving the models' calibration performance?
> - This is the fundamental reason why we introduced the hyperparameter $\eta$ in $\mathcal{L}_{DM(CE)}$. This parameter is the very trade-off between smoothness(calibration) and discrimination(leveraging hard mixed samples). the value of $\eta$ can be referred to as 5.4(1), which generally takes 0.1 by default for static methods and 1.0 for dynamic methods.
>
> > 15. Do you have any insight or principle to determine the "level of necessity" or "effectiveness" of applying DM in a given task and dataset?
> - Although the semantic information is not as centralized in these tasks as in concrete object classification, aggregating some dispersed local features can implicitly construct semantic features with sufficient discriminative information, thus also forming implicit hard mixed samples. Therefore, we have done experiments on Place205 dataset for Decoupled Mixup and show that DM is also gainful for all kinds of mixup algorithms. We evaluate the performance gain of DM(CE) upon various mixup methods based on ResNet-18 on Place205. We follow the settings of the Place205 mixup benchmark in OpenMixup. The results are shown in the following table. We can thus conclude that our proposed DM(CE) loss can also mine hard mixed samples in the scenic tasks.
>
> | Methods     | $\alpha$ | (MCE) | +DM(CE) |
> |-------------|----------|-------|---------|
> | MixUp       | 0.2      | 59.33 | +0.38   |
> | CutMix      | 0.2      | 59.21 | +0.49   |
> | ManifoldMix | 0.2      | 59.46 | +0.36   |
> | SaliencyMix | 0.2      | 59.50 | +0.27   |
> | FMix        | 0.2      | 59.51 | +0.22   |
> | ResizeMix   | 1        | 59.66 | +0.18   |
> | PuzzleMix   | 1        | 59.62 | +0.19   |
> | AutoMix     | 2        | 59.74 | +0.11   |

---

> > ### Comment · Reviewer_JzuZ · 2023-08-15
> >
> > Thank you for the response. Questions addressed.

---

> > > ### Author Response · Authors · 2023-08-16
> > >
> > > Thank you again for your detailed and high-quality review!

---

### Author Response · Authors · 2023-08-19
**Look forward to post-rebuttal feedback**

Dear Reviewers,

We would like to express our sincere gratitude for dedicating your time to reviewing our manuscript. Your insightful comments and suggestions have been instrumental in refining the quality and clarity of our work. We also thank reviewer JzuZ again for his positive response and discussion participation.

We have thoroughly considered ALL feedback and carefully responded to other reviewers. We hope our responses have addressed your concerns to your satisfaction.

With the improvements made and clarifications provided, we kindly hope you to raise the score for our paper, if you deem fit. If you need any clarification or have any other questions, please do not hesitate to let us know.

Once again, we sincerely thank you for your invaluable contribution to our work and look forward to your post-rebuttal feedback.

Best regards,

Authors.

---

### Decision · Program_Chairs · 2023-09-21

**Decision:**

Accept (poster)

**Comment:**

This work highlights a drawback of the existing $\lambda$ mixup, which suppresses the confidence of the lower weighted class by proposing a decoupled mixup loss. The proposed solution is a simple and elegant loss (decoupled-mixup loss). It involves the removal of the opponent class in the standard softmax. The need for the proposed $\mathcal{L}_{DM}$ is well motivated (Proposition 1)

The authors extensively show results on the supervised and semi-supervised setup and demonstrate its ability to use various existing mixup-based methods. They also show their ability to scale across different architectures ranging from ResNets, WideResNets, and Transformers. The proposed method provides significant savings in computational costs which have been a deterrent in dynamic mixup-based methods that show comparable results.

The authors have taken into account the detailed critique by JzuZ regarding the writing which we expect to be fixed in the camera-ready version of the manuscript.